# mRNA trafficking directs cell-size-scaling of mitochondria distribution and function

Joshua J. Bradbury[1,2], Georgia E. Hulmes [1,4], Ranjith Viswanathan[1,4], Guilherme Costa [1,3], Holly E. Lovegrove[1] & Shane P. Herbert [1] ✉

The subcellular positioning of organelles is critical to their function and is dynamically adapted to changes in cell morphology. Yet, how cells sense shifts in their dimensions and redistribute organelles accordingly remains unclear. Here we reveal that cell-size-scaling of mitochondria distribution and function is directed by polarised trafficking of mRNAs. We identify a 29 bp 3'UTR motif in mRNA encoding TRAK2, a key determinant of mitochondria retrograde transport, that promotes cell-size-dependent targeting of *TRAK2* mRNA to distal sites of cell protrusions. Cell-size-scaled mRNA polarisation in turn scales mitochondria distribution by defining the precise site of TRAK2-MIRO1 retrograde transport complex assembly. Consequently, 3'UTR motif excision perturbs size-regulated transport and eradicates scaling of mitochondria positioning, triggering distal accumulation of mitochondria and progressive hypermotility as cells increase size. Together, our results reveal an RNA-driven mechanistic basis for the cell-size-scaling of organelle distribution and function that is critical to homeostatic control of motile cell behaviour.

The precise regulation of organelle size, number and positioning is fundamental to the control of cell function[1–4]. In the case of mitochondria, tight control of mitochondrial biogenesis, turnover and distribution directs core cellular processes as diverse as cell metabolism, survival and fate determination[5–7]. As cells grow and/or change shape, the demand on organelles shifts, as well as the subcellular positions that they must assume. Consequently, in the face of cell size alterations, cells reduce/expand and/or reposition their organellar pool as appropriate to maintain normal physiological function[1–4]. To achieve this, cells employ a suite of cell-size-scaling rules that robustly scale organelle dynamics with cell size change[1,2]. Yet, despite this phenomenon being a focus of research for over a century[8–10], a mechanistic basis for cell-size-scaling remains largely elusive, especially regarding the cell-size-dependent re-distribution of organelles.

The appropriate subcellular positioning of organelles is largely directed by motor-driven cytoskeletal transport[3,4]. For mitochondria, long-range trafficking in mammalian cells is predominantly mediated by microtubules and driven by either the plus-end directed kinesin-1 or the minus-end directed cytoplasmic dynein-1 (dynein) family of motor proteins[6,11]. These motors are recruited to mitochondria via the trafficking kinesin protein (TRAK)/Milton family of motor-adaptor proteins that connect kinesin-1/dynein to the outer mitochondrial membrane protein, Miro[12–18]. Whilst Milton was first identified in *Drosophila*, its mammalian orthologues TRAK1 and TRAK2 share highly conserved functions in mitochondrial transport. In particular, TRAK1 promotes bidirectional transport via interaction with kinesin-1 and dynein, whereas TRAK2 is thought to predominantly mediate minus-end directed trafficking due to preferential interaction with dynein and its partner complex, dynactin[18–20], although this remains a topic of debate[21,22]. As mitochondria are the powerhouse of cells[23], this cytoskeletal trafficking is fundamental to the spatial control of energy production. Consequently, tight regulation of mitochondria transport, among other functions, drives polarised energy release during cell migration[24–32], fuels local remodelling of neurons[33,34], and determines

[1]Faculty of Biology, Medicine and Health, Michael Smith Building, University of Manchester, Oxford Road, Manchester, UK. [2]Centre for Developmental Neurobiology, New Hunt's House, King's College London, London, UK. [3]School of Medicine, Dentistry and Biomedical Sciences, Wellcome Wolfson Institute for Experimental Medicine, Queen's University Belfast, Belfast, UK. [4]These authors contributed equally: Georgia E. Hulmes, Ranjith Viswanathan. ✉e-mail: shane.herbert@manchester.ac.uk

the fitness of daughter cells following mitosis[35–37]. Moreover, this transport is cell-size-responsive, enabling redistribution of mitochondria as cells shift size[38]. Without such cell-size-scaling mechanisms, even small changes in cell geometry could profoundly impact positioning-dependent mitochondrial functions. Yet, how cells sense their dimensions and adjust mitochondrial trafficking and distribution accordingly remains unclear.

The localisation of mRNAs to distinct subcellular sites is critical to the spatial control of gene expression and function[39]. Of note, we and others recently revealed that mRNA encoding the key mitochondria motor-adaptor, TRAK2, is robustly targeted to distal sites of cell protrusions in diverse cell types[40–42]. Indeed, upon reanalysis of our previous characterisation of protrusion-enriched RNAs[40], we find that *TRAK2* is unique amongst mRNAs encoding mitochondria trafficking machinery (for example, *TRAK1*, *MIRO1* and *MIRO2*), which are not localised to protrusions and display unbiased cellular distributions (Supplementary Fig. 1). To achieve this, *TRAK2* mRNA contains a conserved 3'UTR GA-rich motif, an RNA element that acts to drive RNA-binding protein (RBP)-mediated transport of a suite of mRNAs to the plus-end of microtubules[40,41,43–45]. In contrast, searching for the presence of this RNA motif in *TRAK1* using the Find Individual Motif Analysis (FIMO) programme[46] confirmed that *TRAK1* does not contain any GA-rich motifs. These observations suggest that mRNA localisation modulates a specific but unknown attribute of TRAK2 function. Considering that these GA-rich targeting motifs operate in multiple cell types and are highly conserved between species[47], it is surprising that their function remains largely unexplored. However, recent work is starting to build a consensus view that these GA-rich elements spatially modulate protein-protein interactions by defining the precise site of nascent protein production and co-translational protein complex assembly[45,48,49]. Despite these observations, the function of *TRAK2* mRNA localisation and any role in the control of mitochondrial dynamics remains unknown.

Motile cells exhibit a broad range of cell shape dynamics and adopt distinct morphologies depending on their substrate and mode of migration[50]. For example, primary endothelial cells exhibited a rounded morphology with multiple lamellipodia when migrating on a glass substrate but adopted highly elongated uniaxial shapes on cell-derived matrix (CDM) (Supplementary Fig. 2a–c), similar to endothelial cells in vivo[51]. Consequently, adaptation to substrate composition generates cells that vary widely in both morphometric and migratory characteristics (Supplementary Fig. 2c–e). This broad morphological variation exists even within a cell population on a single substrate, with cells on CDM exhibiting a ninefold variance in aspect ratio, 6-fold differences in circularity and threefold variation in protrusion length (Supplementary Fig. 2c). To deal with this broad diversity in shape, cells must reduce/expand and/or reposition organelles as appropriate to match their geometry[1–4]. Yet, how cells sense their dimensions and act accordingly remains unclear.

Here, using single-molecule mRNA imaging and endogenous gene editing, we show that *TRAK2* mRNA targeting functions to modulate the cell-size-scaling of mitochondria distribution and function. We show that the targeting of *TRAK2* mRNA to distal sites of cell protrusions is itself cell-size-dependent and acts to define the precise site of TRAK2-MIRO1 retrograde transport complex assembly. As such, following broad shifts in cell size, scaled targeting of *TRAK2* mRNA governs cell-size-dependent re-distribution of mitochondria to elegantly maintain appropriate mitochondrial positioning. Consequently, excision of a 29 bp 3'UTR motif in *TRAK2* is alone sufficient to eradicate cell-size-scaling of both mitochondria positioning and cell motility. Taken together, these data define an RNA-mediated mechanistic basis for cell-size-scaling of organelle distribution that directs homoeostatic control of cell migration.

## Results

### TRAK2 mRNA localisation scales with shifts in cell size

We previously identified a cluster of five mRNAs (*TRAK2*, *KIF1C*, *RAB13*, *RASSF3* and *NET1*) that are highly polarised at the leading edge of endothelial cell (EC) protrusions[40]. Using single-molecule RNA imaging, we observed that shifts in cell geometry were tightly associated with changes in the polarised subcellular distribution of these mRNAs (Fig. 1; Supplementary Fig. 3), hinting at a role for mRNA trafficking in cell-size-sensing. In particular, we found that all five mRNAs displayed significant substrate-dependent subcellular polarisation in ECs (Fig. 1a–i; Supplementary Fig. 3a–h). *TRAK2*, *RASSF3*, *RAB13*, *KIF1C* and *NET1* were all consistently diffusely enriched at the leading edge of lamellipodium of cells on glass, however, on CDM, displacement of transcripts at the tip of protrusions appeared even more acute (Fig. 1a, d, g; Supplementary Fig. 3a, d, g, h). Quantification of the position of the mRNA centre of mass (CoM) relative to the nucleus confirmed that all five mRNA populations were displaced further away from the nucleus in cells on CDM versus glass (Fig. 1b, e, h; Supplementary Fig. 3b, e). Indeed, a tight correlation between the positioning of the mRNA CoM and the length of cell protrusions indicated that targeting of these mRNAs scaled with shifts in protrusion size (Fig. 1c, f, i; Supplementary Fig. 3c, f). In contrast, positioning of a diffuse non-targeted control transcript, *GAPDH*, was significantly less impacted by changes in cell morphology (Fig. 1c, f, i; Supplementary Fig. 3c, f). Moreover, *ACTB*, an mRNA that is targeted to protrusions via distinct mechanisms[52], exhibited mRNA CoM values that were indistinguishable from *GAPDH* control (Fig. 1j–l). Thus, active targeting mechanisms underpin cell-size-scaled positioning of *TRAK2*, *RASSF3*, *RAB13*, *KIF1C* and *NET1* mRNAs as cells change shape. Importantly, we also observed that each of these mRNAs exhibited distinct scaling properties (Fig. 1m; Supplementary Fig. 3i). When the position of the RNA CoM was normalised to protrusion length, it was noted that mRNAs encoding *RASSF3* and *RAB13* became increasingly more polarised as protrusions elongate (Fig. 1m). In contrast, localisation of the CoM for *TRAK2* mRNA was remarkably robust to morphological change, being consistently maintained at a position ~60% along the length of cell protrusions, independent of shifts in protrusion size (Fig. 1m, n). As such, whereas the RNA centre of mass for *RASSF3* and *RAB13* is sensitive to shifts in cell size, the precise subcellular distribution of *TRAK2* mRNA is highly robust to changes in size scale. This ability of mRNA trafficking to be cell-size-scaled and demarcate a fixed position in cells of differential length hinted that *TRAK2* mRNA may play a key role in the cellular response to shape change.

### TRAK2 modulates the cell-size-scaling of mitochondria positioning

TRAK2 protein is a well-established cytoskeletal adaptor that modulates motor-dependent mitochondria positioning[18]. Considering that the subcellular positioning of *TRAK2* mRNA was size-dependent (Fig. 1a–c, m, n), we speculated that TRAK2 may be a previously unknown determinant of cell-size-scaled mitochondria positioning. To address this hypothesis, TRAK2 expression was perturbed using siRNAs (Fig. 2a) and the impact on mitochondria positioning assessed following culture of endothelial cells on glass or CDM (Fig. 2b–f). In particular, as TRAK2 is known to promote retrograde transport of mitochondria[18,21,22], we determined if loss of TRAK2 promoted mitochondria accumulation at distal sites, indicative of reduced retrograde movement. Whereas, for cells on glass, we observed no significant differences in the distal localisation of mitochondria (Fig. 2b, c), on CDM, siTRAK2 cells consistently accumulated mitochondria specifically at distal-most sites in protrusions when compared to controls (siCTRL; 0–10 μm region; Fig. 2d–f). This 2-fold enrichment of

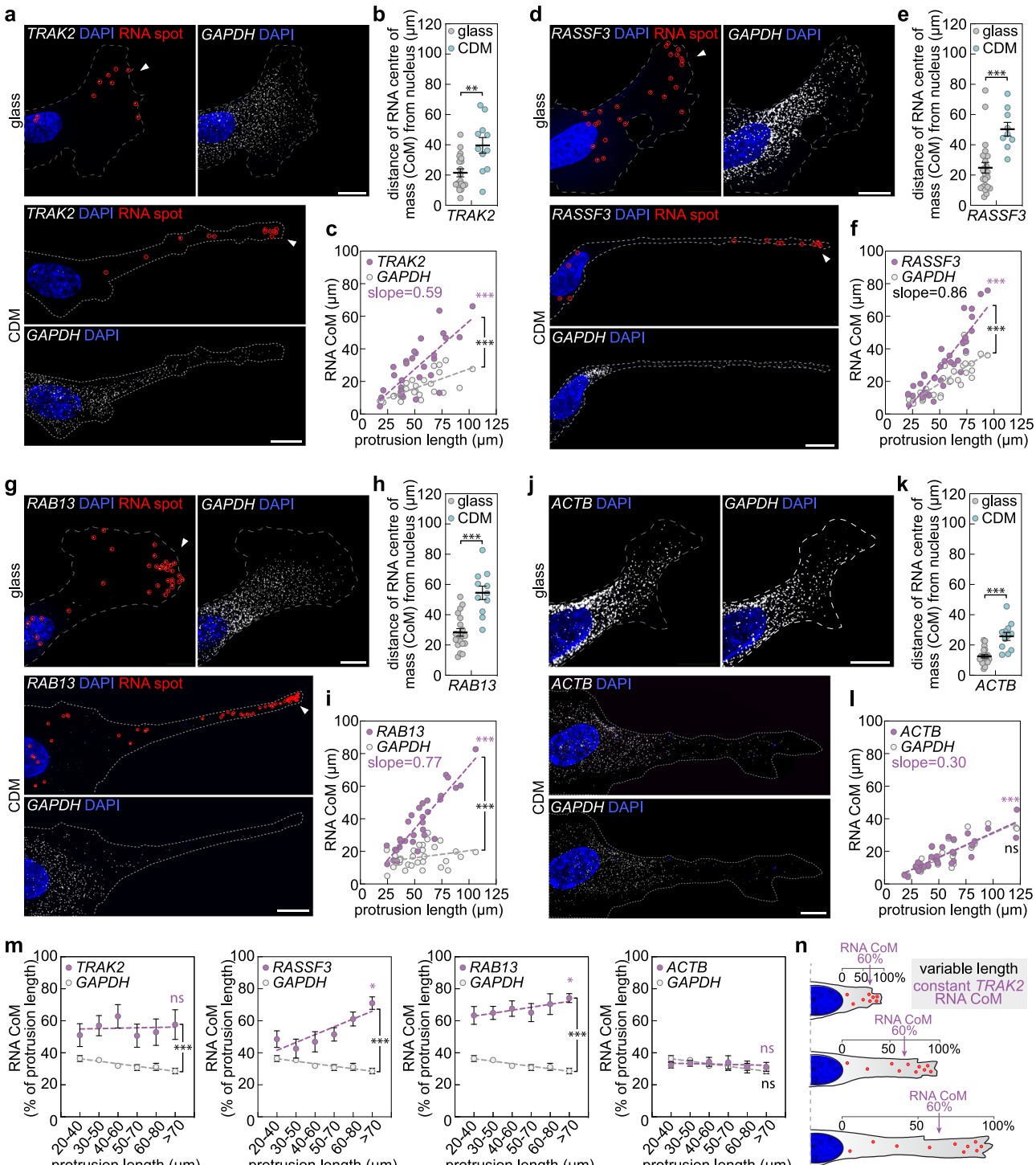

**Fig. 1 | Targeting of *TRAK2* mRNA to cell protrusions scales with shifts in cell size. a, d, g, j** smFISH detection of *TRAK2*, *RASSF3*, *RAB13*, and *ACTB* mRNAs in exemplar ECs cultured either on glass or CDM (red circles indicate distinct mRNA spots; arrowheads indicate mRNA accumulation at distal sites in protrusions; dashed line indicates cell outline). **b, e, h, k** Quantification of the distance that the RNA centre of mass (CoM) sits from the nucleus for the indicated mRNAs when ECs were cultured either on glass or CDM (*n* = 19 cells glass, 11 cells CDM for **b**, *n* = 24 cells glass, 9 cells CDM for **e**, *n* = 20 cells glass, 11 cells CDM for **h**, *n* = 21 cells glass, 12 cells CDM for **k**; two-tailed Mann–Whitney test, \*\*\**P* = 0.0002 for **e** <0.0001 for **h, k, \*\****P* = 0.0053). **c, f, i, l** Plots comparing the distance that the RNA CoM sits from the EC nucleus versus protrusion length for the indicated mRNAs (*n* = 30 cells for **c,** *n* = 33 cells for **f,** *n* = 31 cells for **i,** *n* = 33 cells for **l;** two-tailed Pearson's correlation

coefficient, magenta asterisks, \*\*\**P* ≤ 0.0001 for all; two-sided analysis of covariance, black asterisks, \*\*\**P* = 0.0018 versus *GAPDH* for **c** <0.0001 for **f, i;** ns *P* = 0.8874 versus *GAPDH*). **m** Plots comparing the distance that the RNA CoM sits from the EC nucleus normalised to protrusion length versus protrusion length for the indicated mRNAs (*n* = as for **c, f, i, l;** two-tailed Pearson's correlation coefficient, magenta asterisks, \**P* = 0.0225 for *RASSF3*, 0.0161 for *RAB13*, ns *P* = 0.3506 for *ACTB*, 0.8563 for *TRAK2*; two-sided analysis of covariance, black asterisks, \*\*\**P* = 0.0012 versus *GAPDH* for *RAB13*, <0.0001 versus *GAPDH* for *TRAK2* and *RASSF3*, ns *P* = 0.1304 versus *GAPDH*). **n** Positioning of *TRAK2* mRNA as cells shift in size. For analyses in **c, f, i, l, m** data for cells cultured on glass and CDM were pooled. Data are mean ± s.e.m. (**b, e, h, k, m**). **a, d, g, j** scale bars, 10 μm. Source data is provided as a Source Data file.

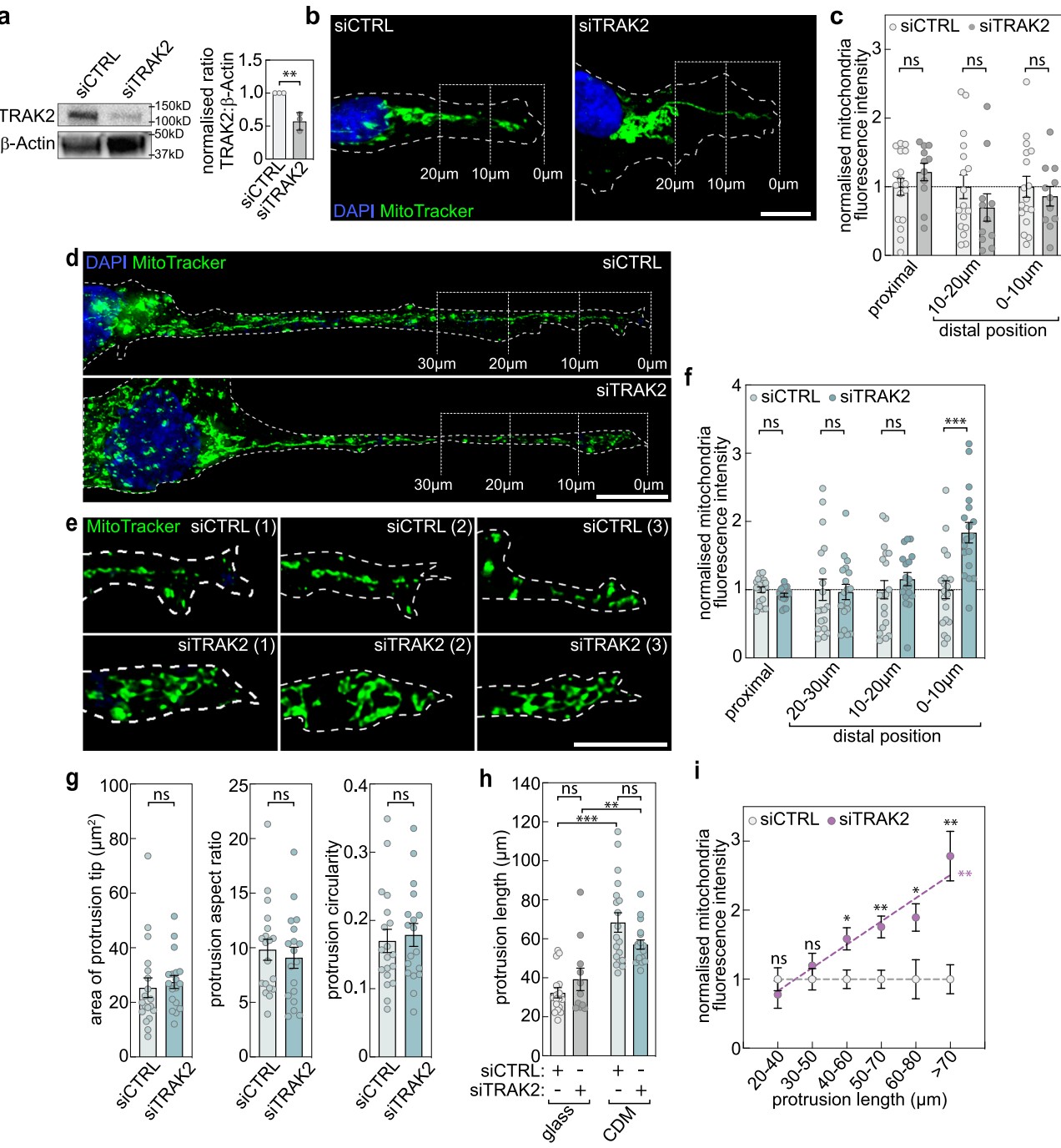

mitochondria in siTRAK2 cells was not indirect due to broad changes in the morphology or length of protrusions (Fig. 2g, h), or perturbation of the microtubule cytoskeleton (Supplementary Fig. 4a, b), indicating that TRAK2 is indeed required for retrograde mitochondria trafficking. However, the presence of this phenotype only in siTRAK2 cells cultured on CDM, which are longer than cells cultured on glass (Supplementary Fig. 2c), suggested that TRAK2-mediated mitochondria transport was highly sensitive to the uniaxial elongation of cells. Indeed, quantification of the enrichment of mitochondria at the distal-most leading edge of siTRAK2 cells revealed a clear size-dependent progressive accumulation of mitochondria as protrusions increase in length (Fig. 2i). Together, these data confirm that TRAK2 influences cell-size-scaling of mitochondria spatial positioning, acting to maintain an appropriate subcellular distribution as cells change morphology.

## A 29 bp motif directs cell-size-scaling of *TRAK2* mRNA distribution

To determine if the cell-size-dependent trafficking of *TRAK2* mRNA mechanistically underpins TRAK2-mediated size-scaling of mitochondria distribution, we aimed to specifically manipulate mRNA localisation. We and others previously identified a conserved GA-rich 3'UTR motif element that is necessary and sufficient to drive localisation of mRNAs to distal sites of cell protrusions[40,45]. Such localisation elements are often repeated multiple times in 3'UTRs, with *TRAK2* containing three such motif repeats[40]. Hence, we first identified which of these GA-rich motifs was critical to the targeting of *TRAK2* mRNA to protrusions. To achieve this, we used the MS2-MCP reporter construct system to expand our previous analysis of the *TRAK2* 3'UTR[40]. With this tool, a non-targeted *HBB* coding sequence was tagged with 24x bacteriophage-derived MS2 hairpin repeats, which act as a high-affinity

**Fig. 2 | TRAK2 modulates the cell-size-scaling of mitochondria positioning.**
**a** Representative western blot for TRAK2 and β-actin in ECs treated with either control (siCTRL) or *TRAK2*-targeting (siTRAK2) siRNAs and densitometric analysis of the ratio of TRAK2:β-Actin levels (*n* = 3 independent experiments, two-tailed unpaired *t* test, **P = 0.0051). **b** Representative images of MitoTracker-labelled mitochondria in siCTRL- or siTRAK2-treated ECs cultured on glass (brackets indicate the distal-most 10 μm and 20 μm of protrusions; dashed line indicates cell outline as defined by F-actin labelling). **c** Quantification of mitochondria levels at the indicated positions of EC protrusions following culture on glass and treatment with siCTRL or siTRAK2 (data are normalised to siCTRL, *n* = 17 cells siCTRL, *n* = 11 cells siTRAK2, two-tailed Mann–Whitney test, ns *P* = > 0.2441). **d**, **e** Representative images of MitoTracker-labelled mitochondria in whole protrusions (**d**) or the distal-most 10 μm of protrusions (**e**) of siCTRL- or siTRAK2-treated ECs cultured on CDM (brackets indicate the distal-most 10 μm, 20 μm, and 30 μm of protrusions; dashed line indicates cell outline as defined by F-actin labelling). **f** Quantification of mito-chondria levels at the indicated positions of EC protrusions following culture on CDM and treatment with siCTRL or siTRAK2 (data are normalised to siCTRL, *n* = 19

cells siCTRL, *n* = 18 cells siTRAK2, unpaired Kruskal–Wallis test and Dunn multiple comparison test, ***P = 0.0003, ns *P* ≥ 0.6664). **g** Quantification of protrusion morphometrics (area, aspect ratio, and circularity) of ECs cultured on CDM and treated with siCTRL or siTRAK2 (*n* = 19 cells siCTRL, *n* = 18 cells siTRAK2, two-tailed Mann–Whitney test, ns *P* ≥ 0.2844). **h** Quantification of EC protrusion length following culture on glass or CDM and treatment with siCTRL or siTRAK2 (*n* = 17 cells siCTRL glass, *n* = 11 cells siTRAK2 glass, *n* = 19 cells siCTRL CDM, *n* = 18 cells siTRAK2 CDM, two-tailed Mann–Whitney test, **P = 0.0018, ***P ≤ 0.0001, ns *P* = 0.2100 for CDM siCTRL vs CDM siTRAK2, 0.4581 for glass siCTRL vs glass siTRAK2). **i** Plot comparing mitochondria levels at the indicated positions of EC protrusions following treatment with siCTRL or siTRAK2 (data normalised to siCTRL, *n* = 36 cells siCTRL, *n* = 29 cells siTRAK2, two-tailed unpaired *t* test, *P = 0.0125 for 40–60 μm, 0.0329 for 60–80 μm, **P ≤ 0.0023 for 50–70 μm, 0.001 for >70 μm, ns *P* = 0.4211 for 20–40 μm, 0.4112 for 30–50 μm; two-sided analysis of covariance, **P = 0.002 versus siCTRL). For analyses in **i**, data for cells cultured on glass and CDM were pooled. Data are mean ± s.e.m. (**a**, **c**, **f–i**). For **b**, **d** scale bars, 10 μm. **e** Scale bar, 5 μm. Source data are provided as a Source Data file.

binding site for the MS2-capping protein (MCP). Co-expression of MCP-nlsGFP can then be used to visualise the subcellular localisation of MS2-tagged RNAs. Fusion of the *HBB*-MS2 construct with wild-type (Wt) or motif-excised *TRAK2* 3'UTRs thus enabled the function of each motif to be determined (Supplementary Fig. 5a). In controls lacking a 3'UTR fusion, the nuclear localisation tag of MCP-nlsGFP restricted localisation to the nucleus (Supplementary Fig. 5b, c). In contrast, fusion with the Wt *TRAK2* 3'UTR, a truncated 3'UTR containing all three G-motifs (1-1280 bp), or a truncated 3'UTR lacking the central 374-403bp G-motif all promoted robust GFP accumulation at distal sites of cell protrusions (Supplementary Fig. 5b, c). However, removal of just the 114–143 bp proximal-most G-motif was sufficient to eradicate protrusion-targeting properties of the 3'UTR (Supplementary Fig. 5c). Hence, this single 29bp G-motif acted as a critical localisation element driving polarised mRNA trafficking.

To assess the endogenous function of this 29 bp motif, CRISPR-Cas9 was used to excise the 114–143 bp region from the endogenous *TRAK2* 3'UTR. Guide RNAs were designed to flank the 29bp G-motif (Fig. 3a) and CRISPR-Cas9 ribonucleoprotein complexes nucleofected into endothelial cells alongside a GFP expression vector. Single GFP-positive cells were isolated, expanded, sequenced, and subjected to PCR to identify clones containing homozygous G-motif deletion (*ΔTRAK2*; Fig. 3a, b; Supplementary Fig. 6a, b). Control Wt clones were also created upon nucleofection of ribonucleoprotein complexes without guide RNA (Supplementary Fig. 6a). Single-molecule imaging of endogenous mRNAs in cells on glass revealed that loss of the G-motif efficiently eradicated polarised targeting of *TRAK2* to protru-sions (Supplementary Fig. 6c–e). *TRAK2* mRNA was visibly depolarised and more proximally distributed in *ΔTRAK2* versus Wt cells (Supple-mentary Fig. 6c). Moreover, the polarisation index[53] of *TRAK2* mRNA was significantly lower in *ΔTRAK2* cells, and became indistinguishable from the diffuse control transcript, *GAPDH* (Supplementary Fig. 6d). Likewise, *TRAK2* mRNA dispersion index[53] was also significantly lower in *ΔTRAK2* cells (Supplementary Fig. 6e). In contrast, *GAPDH* mRNA polarisation and dispersion were unchanged (Supplementary Fig. 6c–e). Importantly, excision of the G-motif also had no impact on *TRAK2* mRNA spot or TRAK2 protein levels, indicating that loss of polarised mRNA transport did not perturb mRNA/protein turnover dynamics (Supplementary Fig. 6f, g). Thus, the 29bp G-motif is a cri-tical modulator of *TRAK2* spatial distribution that promotes polarised mRNA trafficking to distal sites of cell protrusions.

Similar to cells cultured on glass, we noted that *TRAK2* mRNA was also significantly depolarised in *ΔTRAK2* cells cultured on CDM (Fig. 3c, d). Again, excision of the G-motif had no impact on total mRNA spot levels (Fig. 3e), however, polarised enrichment of *TRAK2* spots at the distal-most 0–10 μm leading edge of cell protrusions was entirely lost upon G-motif excision (Fig. 3c, d, f). Consequently, *TRAK2* mRNA was

significantly more proximally located in *ΔTRAK2* cells (Fig. 3d, f), and instead resembled the non-polarised distribution of the diffuse control mRNA, *GAPDH* (Fig. 3g). Likewise, quantification of the position of the mRNA CoM relative to the nucleus in Wt and *ΔTRAK2* cells also con-firmed that mutant *TRAK2* mRNA was significantly less polarised and was positioned closer to the nucleus, being indistinguishable from *GAPDH* controls (Fig. 3h). Loss of the G-motif had no impact on the length of cell protrusions, indicating that *TRAK2* mRNA distribution does not impact cell size control and that loss of mRNA polarisation was not an indirect consequence of any reduction in protrusion length (Fig. 3i). Thus, when the position of the *TRAK2* mRNA CoM was nor-malised to protrusion length, the CoM in *ΔTRAK2* cells was no longer maintained at ~60% along the length of cell protrusions, was much more proximally localised and again was indistinguishable from *GAPDH* controls (Fig. 3j). In contrast to *TRAK2*, positioning of the CoM of another G-motif-containing mRNA, *RAB13*, was unperturbed in *ΔTRAK2* cells, confirming specificity of this approach (Supplementary Fig. 6h). Most importantly, G-motif excision indeed significantly per-turbed the cell-size-scaling properties of *TRAK2* mRNA distribution. In particular, the robust correlation between positioning of the mRNA CoM and the length of cell protrusions was significantly attenuated in *ΔTRAK2* cells, to now resemble *GAPDH* controls (Fig. 3k). Moreover, this resulted in a significant difference in *TRAK2* mRNA distribution when protrusions extended beyond ~60 μm in length in *ΔTRAK2* cells (Fig. 3l). Overall, these data reveal that the 29 bp 3'UTR G-motif underpins cell-size-dependent positioning of *TRAK2* mRNA, and that the impact of motif excision was progressively more severe as pro-trusions elongated past a threshold length of ~60 μm.

## *TRAK2* mRNA directs the cell-size-scaling of mitochondria distribution

Next, we determined if the loss of cell-size-dependent *TRAK2* mRNA distribution had any impact on mitochondria positioning. First, we observed that excision of the G-motif was itself sufficient to pheno-copy *TRAK2* knockdown (Fig. 2d–f), resulting in robust accumulation of mitochondria at the distal-most sites of *ΔTRAK2* cell protrusions when compared to Wt control cells on CDM (0–10 μm region; Fig. 4a–c). Again, this near twofold enrichment of mitochondria was not indirect due to broad changes in the morphology or length of cell protrusions on CDM, which remained unchanged (Fig. 4d). Likewise, the microtubule cytoskeleton was not perturbed in *ΔTRAK2* cells (Supplementary Fig. 7a, b). Moreover, further analysis revealed a clear interrelationship between *TRAK2* mRNA and mitochondria positioning (Fig. 4e). In Wt cells, an increase in *TRAK2* mRNA spot numbers at the distal sites of cell protrusions was coupled with a decrease in mito-chondria enrichment at these sites. In contrast, absence of *TRAK2* mRNAs at distal sites in *ΔTRAK2* cell protrusions was linked to elevated

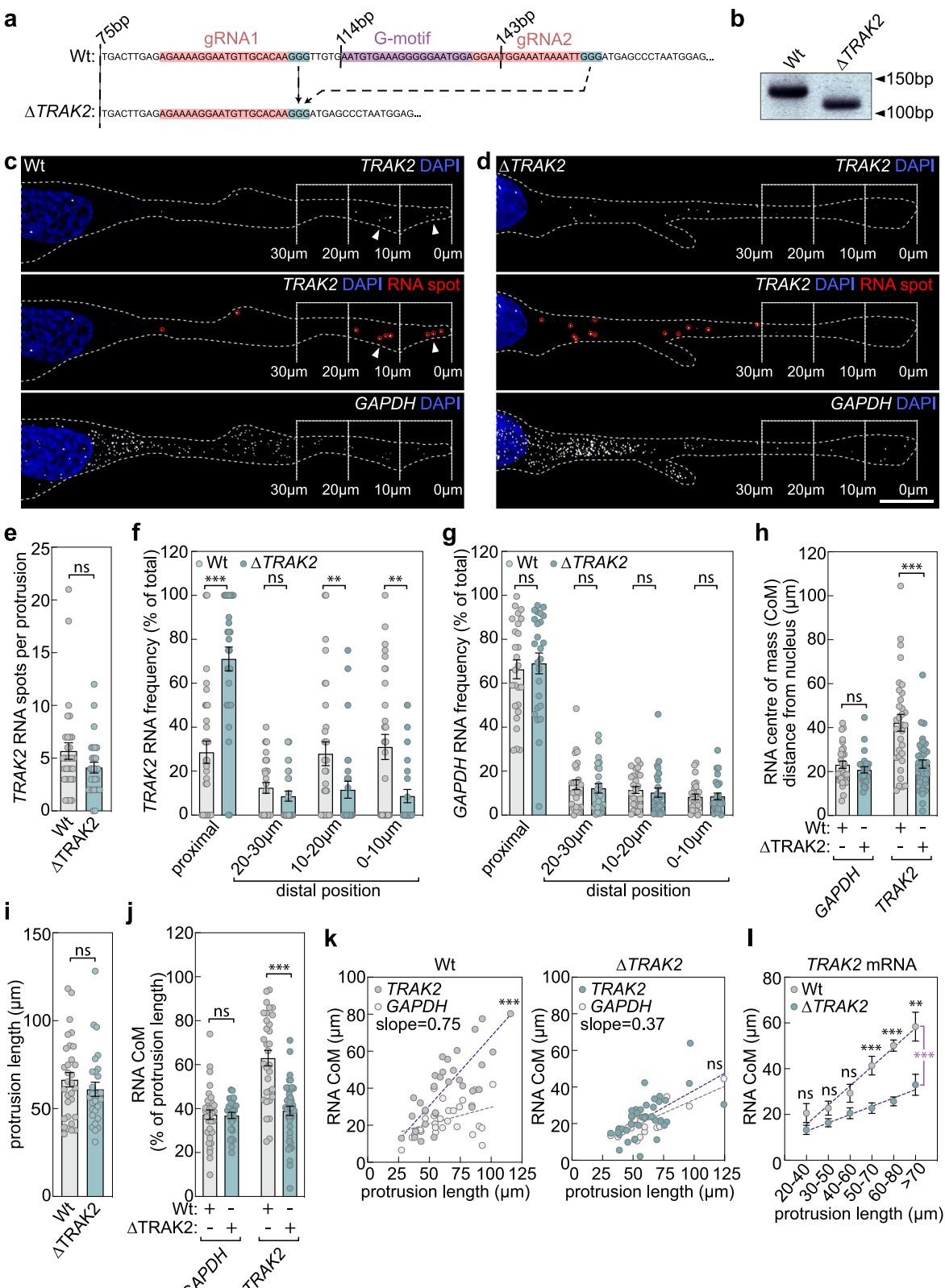

mitochondria enrichment relative to Wt controls (Fig. 4e). Similar to the knockdown of *TRAK2* (Fig. 2b–f), this mis-localisation of mito-chondria was specific to *ΔTRAK2* cells cultured on CDM, as no differences in mitochondria enrichment were observed in cells on glass (Fig. 4f), again suggesting a cell-size-dependent phenotype. Indeed, quantification of the enrichment of mitochondria at the distal-most leading edge of *ΔTRAK2* cells confirmed size-dependent progressive

accumulation of mitochondria as protrusions increase in length (Fig. 4g). Strikingly, a significant ectopic accumulation of mitochondria was only observed in cells with protrusions longer than ~60 μm in length (Fig. 4g), mirroring the size threshold at which *TRAK2* mRNA distribution is significantly perturbed in *ΔTRAK2* cells (Fig. 3l). Furthermore, if cells cultured on CDM were subdivided into populations with either short (<60 μm; Fig. 4h) or long (>60 μm; Fig. 4i) protrusions,

**Fig. 3 | A 29 bp 3'UTR motif directs cell-size-scaling of *TRAK2* mRNA distribution. a** Sequences of *TRAK2* 3'UTRs from Wt and *ΔTRAK2* ECs indicating the location of gRNA target (red), NGG PAM (blue), and G-motif sequences (magenta). **b** PCR of the *TRAK2* 3'UTR demonstrating a band shift in *ΔTRAK2* ECs (*n* = 3 experimental repeats). **c, d** smFISH co-detection of *TRAK2* and *GAPDH* mRNAs in Wt (**c**) or *ΔTRAK2* (**d**) ECs (red circles indicate mRNA spots; arrowheads indicate distal mRNA accumulation; brackets indicate distal-most 10 μm, 20 μm, and 30 μm of protrusions; dashed line indicates cell outline). **e** Number of *TRAK2* mRNA spots in Wt and *ΔTRAK2* ECs (*n* = 31 cells Wt, *n* = 28 cells *ΔTRAK2*, two-tailed Mann–Whitney test, ns *P* = 0.1130). **f, g** Relative distribution of *TRAK2* mRNAs (**f**) or *GAPDH* mRNAs (**g**) in Wt and *ΔTRAK2* EC protrusions (*n* = 31 cells Wt *TRAK2*, *n* = 28 cells *ΔTRAK2 TRAK2*, *n* = 26 cells Wt *GAPDH*, *n* = 25 cells *ΔTRAK2 GAPDH*, two-tailed Mann–Whitney test, ***\*P* ≤ 0.0001, **\*\*P* ≤ 0.0096 for 10–20 μm in **f**, 0.0044 for 0–10 μm in **f**, ns *P* = 0.2848 in **f**, 0.591 for proximal in **g**, 0.556 for 20–30 μm in **g**, 0.3945 for 10–20 μm in **g**, 0.8702 for 0–10 μm in **g**). **h** Distance the RNA CoM sits from the nucleus for *GAPDH* and *TRAK2* mRNAs in Wt and *ΔTRAK2* ECs (*n* = 29 cells Wt *GAPDH*, 26 cells *ΔTRAK2 GAPDH*, *n* = 32 cells Wt *TRAK2*, 37 cells *ΔTRAK2 TRAK2*, two-tailed Mann–Whitney test, ***\*P* ≤ 0.0001, ns *P* = 0.2793). **i** Protrusion length in Wt and *ΔTRAK2* ECs (*n* = 31 cells Wt, *n* = 28 cells *ΔTRAK2*, two-tailed Mann–Whitney test, ns *P* = 0.3282). **j** Distance the RNA CoM sits from the nucleus normalised to protrusion length for *GAPDH* and *TRAK2* mRNAs in Wt and *ΔTRAK2* ECs (*n* = 33 cells Wt *GAPDH*, 30 cells *ΔTRAK2 GAPDH*, 32 cells Wt *TRAK2*, 37 cells *ΔTRAK2 TRAK2*, two-tailed unpaired *t* test, ***\*P* ≤ 0.0001, ns *P* = 0.8949). **k** Plots comparing the distance that the RNA CoM sits from the EC nucleus versus protrusion length for *GAPDH* and *TRAK2* mRNAs in either Wt or *ΔTRAK2* ECs (*n* = at least 26 cells, two-sided analysis of covariance, ***\*P* ≤ 0.0001 versus *GAPDH*, ns *P* = 0.3410 versus *GAPDH*). **l** Plot comparing the distance that the *TRAK2* RNA CoM sits from the nucleus in Wt and *ΔTRAK2* ECs with protrusion length (*n* = 32 cells Wt, *n* = 37 cells *ΔTRAK2*, two-tailed unpaired *t* test, ***\*P* = 0.0002 for 50–70 μm, <0.0001 for 60–80 μm, **\*\*P* = 0.0067, ns *P* = 0.1523 for 20–40 μm, 0.0753 for 30–40 μm, 0.0530 for 40–60 μm Wt versus *ΔTRAK2*; two-sided analysis of covariance, magenta asterisk, ***\*P* = 0.0005 Wt versus *ΔTRAK2*). All cells cultured on CDM except **k**, **l** where data for cells cultured on glass and CDM were pooled. Data are mean ± s.e.m. (**e**–**j**, **l**). **c, d** Scale bar, 10 μm. Source data are provided as a Source Data file.

---

significant ectopic accumulation of mitochondria was only observed in *ΔTRAK2* mutant cells with the longest protrusions (Fig. 4i). Consequently, size-scaled *TRAK2* mRNA trafficking critically directs the cell-size-dependent positioning of mitochondria during cell migration, with excision of the *TRAK2* G-motif triggering a near threefold accumulation of mitochondria at the leading edge of elongated cells (Fig. 4i).

### *TRAK2* mRNA trafficking modulates cell-size-scaling of cell motility

Considering that a single G-motif in the 3'UTR of *TRAK2* modulated the spatial distribution of mitochondria, we aimed to define the functional significance of this observation. Tight control of mitochondria spatial positioning at the leading edge of migrating cells is critical to the control of motile cell behaviour[24–32]. As such, we hypothesised that ectopic accumulation of mitochondria at distal sites of *ΔTRAK2* cells may increase local mitochondrial activity. To test this, we quantified mitochondrial membrane potential upon staining of Wt and *ΔTRAK2* cells with the cell-permeant dye, tetramethylrhodamine methyl ester (TMRM)[30]. Consistent with increased mitochondrial function at the leading edge, we observed that the ratio of TMRM to mitochondria staining was significantly enhanced at distal sites of cell protrusions in *ΔTRAK2* versus Wt cells (Fig. 5a–c). Thus, not only did mitochondria accumulate at distal regions of *ΔTRAK2* cells, but these mitochondria also exhibited enhanced activity versus those in Wt cells. Considering the key role of local mitochondria activity in driving cell migration[24–32], we predicted that this ectopic accumulation of active mitochondria at distal sites may deregulate the migration of cells as they change shape. To explore this, we quantified key motile characteristics upon live imaging of Wt and *ΔTRAK2* cells cultured on CDM. Consistent with a key role for distally localised mitochondria in the production of ATP that fuels migration, *ΔTRAK2* cells that accumulated mitochondria at their leading edge exhibited a marked increase in velocity, as well as the accumulated distance and Euclidean distance migrated versus Wt controls (Fig. 5d–g). In contrast, the directionality of *ΔTRAK2* cells remained equivalent to Wt cells (Fig. 5h), indicating an increase in inherent motility, but not directional sensing, of migrating cells upon excision of the *TRAK2* G-motif. Importantly, this enhanced motility of *ΔTRAK2* cells was again cell-size-dependent (Fig. 5i). When Wt control populations were subdivided into cells with either short (<60 μm average length) or long (>60 μm average length) protrusions, cells exhibited equivalent motility independent of any differences in size. However, when *ΔTRAK2* populations were likewise subdivided, cells with short protrusions (<60 μm average length) displayed motility identical to Wt controls, whereas it was exclusively the cells with long protrusions (>60 μm average length) that were significantly more motile (Fig. 5i). Further analysis revealed that the velocity of Wt cells

was remarkably robust to broad changes in cell length (Fig. 5j). In stark contrast, the motility of *ΔTRAK2* cells was no longer robust to shifts in cell size, instead being significantly correlated with changes in protrusion length (Fig. 5k). Comparison of cell motility rates further confirmed that Wt cells exhibited remarkable homoeostatic control of cell velocity, maintaining a constant rate of motility independent of shifting cell size (Fig. 5l). However, upon excision of the *TRAK2* G-motif, this homoeostatic control was lost, with *ΔTRAK2* cells displaying progressive hypermotility as they increased in size (Fig. 5l). Importantly, a significant increase in the motility of *ΔTRAK2* versus Wt cells was only observed in cells with protrusions longer than ~60 μm in length (Fig. 5l), again mirroring the size threshold at which cell-size-dependent *TRAK2* mRNA and mitochondria distribution are significantly perturbed upon G-motif excision (Figs. 3l and 4g). Together, these data revealed that cell-size-scaled *TRAK2* mRNA distribution acts as a previously unknown homoeostatic control mechanism that maintains a consistent rate of cell motility in the face of dynamic changes in cell size.

### *TRAK2* RNA localisation spatially modulates TRAK2-MIRO1 interaction

Whilst data presented above confirmed that *TRAK2* mRNA distribution fundamentally modulates cell-size-scaling of both mitochondria distribution and cell motility, a mechanistic basis for these observations remained unclear. However, recent work suggests that G-motif-mediated subcellular targeting of mRNAs defines the precise site of nascent protein production to modulate co-translational protein complex assembly and/or association of newly translated protein with specific interaction partners[45,48,49]. Considering that binding of TRAK2 to the mitochondrial membrane protein, MIRO1, is critical to dynein-mediated retrograde mitochondria transport[12–22] (Fig. 6a), we tested whether *TRAK2* mRNA polarisation modulated the association of TRAK2 with MIRO1. Contrary to a role for *TRAK2* mRNA targeting in defining the frequency of TRAK2-MIRO1 interactions, co-immunoprecipitation studies revealed that excision of the *TRAK2* G-motif had no impact on association of TRAK2 with MIRO1 (Fig. 6b). However, use of proximity ligation assays (PLA) to define the precise site of close TRAK2-MIRO1 interaction (i.e., both proteins being <40 nm apart) revealed that *TRAK2* mRNA localisation acts to precisely spatially modulate the association of TRAK2 with MIRO1 (Fig. 6c–g). As expected, PLA signal was not detected in control cells lacking either primary antibody against TRAK2 or MIRO1 (Fig. 6c). In comparison, PLA using antibodies against both TRAK2 and MIRO1 uncovered that interaction of these proteins was present all along the length of Wt cells (Fig. 6c). Quantification confirmed that TRAK2-MIRO1 association was not biased to any position and was equally distributed across the length Wt cells (Fig. 6d). In contrast, PLA revealed that excision of the

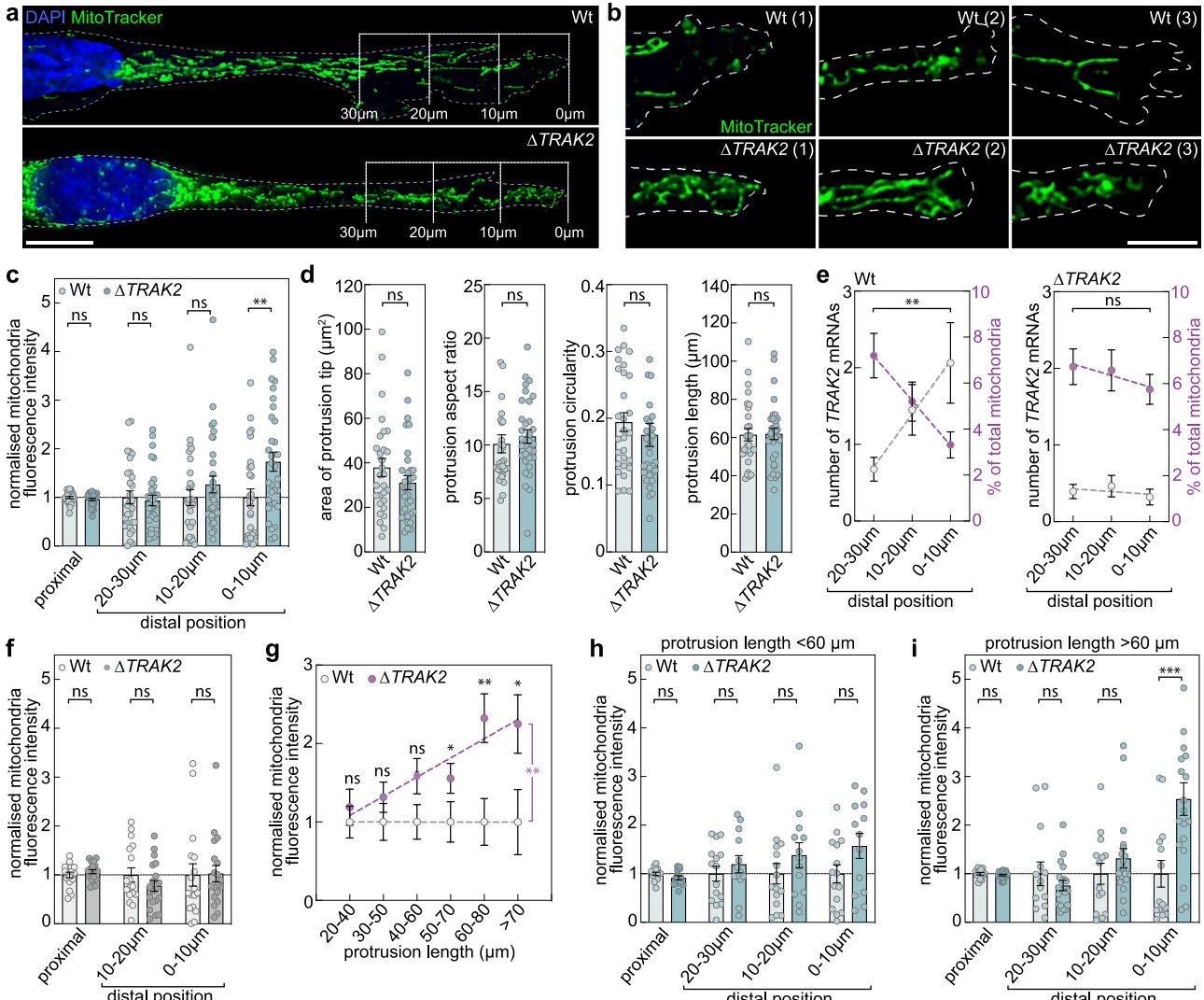

**Fig. 4 | *TRAK2* mRNA trafficking directs cell-size-scaling of mitochondria distribution. a, b** Images of MitoTracker-labelled mitochondria in whole protrusions (**a**) or the distal-most 10 μm of protrusions (**b**) of Wt or *ΔTRAK2* ECs cultured on CDM (brackets indicate distal-most 10 μm, 20 μm and 30 μm of protrusions; dashed line indicates cell outline). **c** Mitochondria levels at the indicated positions of Wt or *ΔTRAK2* EC protrusions following culture on CDM (data is normalised to Wt, *n* = 29 cells Wt, *n* = 32 cells *ΔTRAK2*, unpaired Kruskal–Wallis test and Dunn multiple comparison test, **P = 0.0042, ns P ≥ 0.9999). **d** Quantification of protrusion morphometrics (area, aspect ratio, circularity, and protrusion length) of Wt or *ΔTRAK2* ECs cultured on CDM (*n* = 29 cells Wt, *n* = 32 cells *ΔTRAK2*, two-tailed Mann–Whitney test, ns P ≥ 0.1109). **e** Plots comparing the number of *TRAK2* mRNA spots (grey data points and dashed line) and mitochondria levels (magenta data points and dashed line) at the indicated distal-most positions of Wt or *ΔTRAK2* EC protrusions (*n* = 31 cells Wt *TRAK2*, 28 cells *ΔTRAK2 TRAK2*, 29 cells Wt mitochondria, 32 cells *ΔTRAK2* mitochondria, two-tailed Mann–Whitney test, **P = 0.0030 for % of total mitochondria in Wt, ns P = 0.4115 for % of total mitochondria in *ΔTRAK2*).

**f** Mitochondria levels at the indicated positions of Wt or *ΔTRAK2* EC protrusions following culture on glass (data is normalised to Wt, *n* = 17 cells Wt, *n* = 19 cells *ΔTRAK2*, two-tailed Mann–Whitney test, ns P ≥ 0.2711). **g** Plots comparing mitochondria levels in the distal-most 10 μm of protrusions in Wt or *ΔTRAK2* ECs with protrusions of the indicated length (data is normalised to Wt, *n* = 48 cells Wt, *n* = 46 cells *ΔTRAK2*, two-tailed Mann–Whitney test, *P = 0.0494 for 50–70 μm, 0.0293 for >70 μm, **P = 0.0067, ns P = 0.3426 for 20–40 μm, 0.0919 for 30–50 μm, 0.0786 for 40–60 μm; two-sided analysis of covariance, magenta asterisk, **P = 0.0083 Wt versus *ΔTRAK2*). For analyses in (**g**) data for cells cultured on glass and CDM were pooled. **h, i** Mitochondria levels in protrusions of Wt or *ΔTRAK2* ECs with protrusions either <60 μm in length (**h**) or >60 μm in length (**i**) and following culture on CDM (data is normalised to Wt, *n* = 15 cells Wt <60 μm, *n* = 13 cells *ΔTRAK2* < 60 μm, *n* = 14 cells Wt >60 μm, *n* = 19 cells *ΔTRAK2* > 60 μm, unpaired Kruskal–Wallis test and Dunn multiple comparison test, ***P = 0.0009, ns P = > 0.4018). Data are mean ± s.e.m. (**c–i**). **a** scale bars, 10 μm. **b** scale bar, 7.5 μm. Source data are provided as a Source Data file.

*TRAK2* G-motif significantly displaced the positioning of TRAK2-MIRO1 interactions, predominantly restricting them to the proximal-most aspect of migrating *ΔTRAK2* cells (Fig. 6c, e). Changes in PLA foci positioning were not due any indirect effect on cell protrusion length (Fig. 6f) and were driven by a significant re-distribution of TRAK2-MIRO1 associations towards the cell body (Fig. 6g). Moreover, consistent with no impact on the frequency of interaction between TRAK2 and MIRO1, there was no difference in the total number of PLA foci detected in Wt versus *ΔTRAK2* cells (Fig. 6h). Hence, we propose that *TRAK2* mRNA targeting to the leading edge of migrating cells functions

to specifically promote distal interactions of TRAK2 and MIRO1, ensuring robust cell-size-scaled retrograde transport of mitochondria and homoeostatic control of motile behaviour. In comparison, excision of the *TRAK2* G-motif almost appears to exclusively restrict TRAK2-MIRO1 association to the proximal aspect of migrating cells, leading to distal accumulation of mitochondria and progressive ectopic hypermotility as cells increase size (Fig. 6i). Taken together, these results define an RNA-based mechanistic principle for cell-size-dependent regulation of both mitochondria distribution and cell migration, which has fundamental implications for our understanding

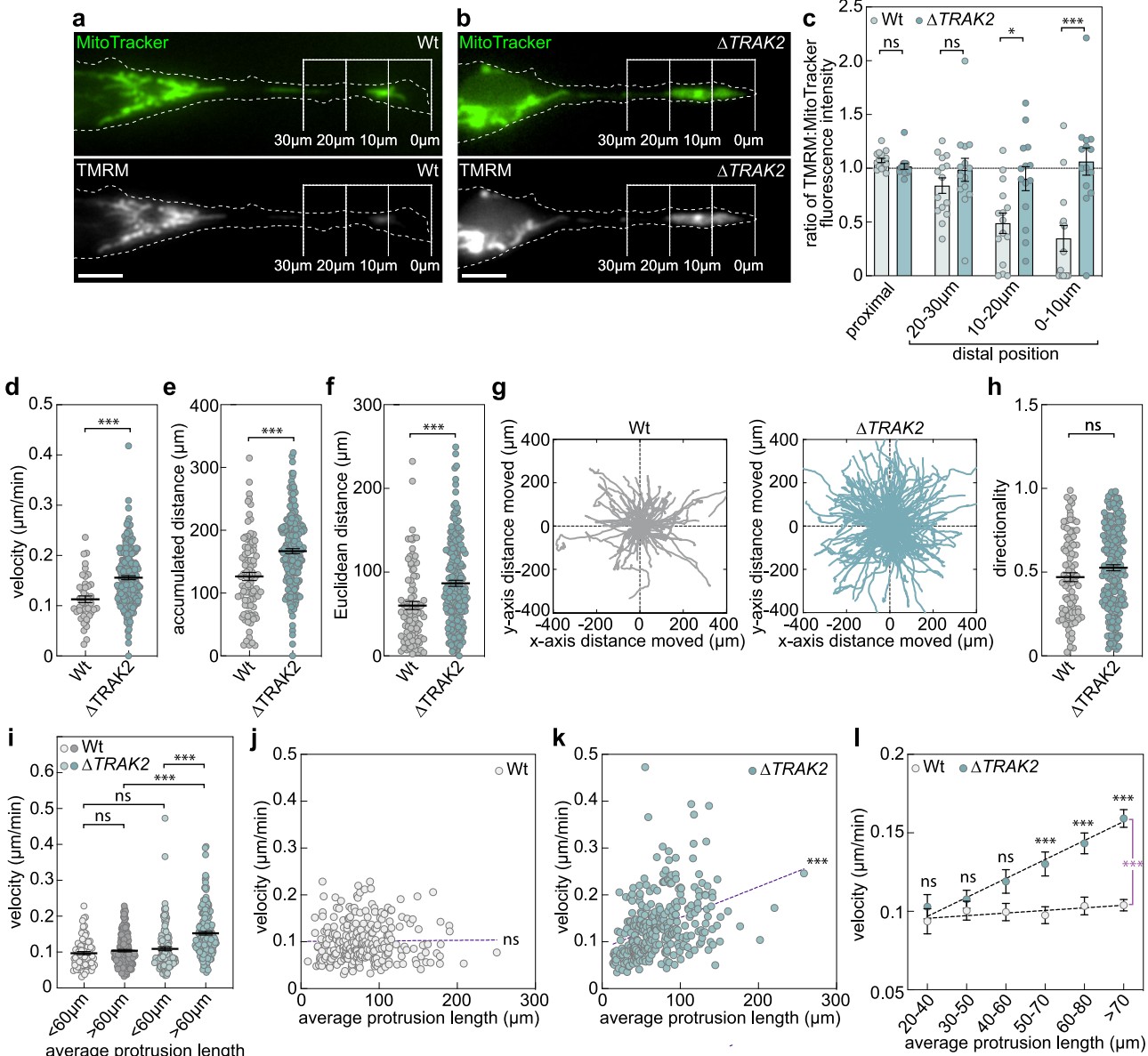

**Fig. 5 | *TRAK2* mRNA trafficking modulates cell-size scaling of cell motility. a, b** Representative images of MitoTracker- and TMRM-labelled mitochondria in protrusions of Wt (**a**) or *ΔTRAK2* (**b**) ECs (brackets indicate the distal-most 10 μm, 20 μm and 30 μm of protrusions; dashed line indicates cell outline as defined by wide-field imaging). **c** Quantification of the ratio of TMRM to MitoTracker staining levels at the indicated positions of Wt or *ΔTRAK2* EC protrusions (*n* = 14 cells Wt, *n* = 15 cells *ΔTRAK2*, unpaired Kruskal–Wallis test and Dunn multiple comparison test, \**P* = 0.0411, \*\*\**P* = 0.0003, ns *P* ≥ 0.9999). **d–f** Quantification of the velocity (**d**, *n* = 45 cells Wt, 209 cells *ΔTRAK2*), accumulated distance (**e**, *n* = 91 cells Wt, 209 cells *ΔTRAK2*), Euclidian distance (**f**, *n* = 91 cells Wt, 209 cells *ΔTRAK2*) of Wt and *ΔTRAK2* ECs cultured on CDM (two-tailed Mann–Whitney test, \*\*\**P* ≤ 0.0001 for all). **g**, Rose plots of Wt and *ΔTRAK2* motile EC trajectories when cultured on CDM (*n* = 91 cells Wt, *n* = 208 cells *ΔTRAK2*). **h** Quantification of the directionality of Wt and *ΔTRAK2* ECs cultured on CDM (*n* = 91 cells Wt, *n* = 208 cells *ΔTRAK2*, two-tailed Mann–Whitney test, ns *P* = 0.0704). **i** Quantification of the velocity of non-

elongated (<60 μm protrusions) and elongated (>60 μm protrusions) Wt and *ΔTRAK2* ECs cultured on CDM (*n* = 80 cells Wt <60 μm, *n* = 171 cells Wt >60 μm, *n* = 130 cells *ΔTRAK2* < 60 μm, *n* = 180 cells *ΔTRAK2* > 60 μm, Mann–Whitney test, \*\*\**P* ≤ 0.0001 for all, ns *P* = 0.2786 for Wt <60 μm vs Wt >60 μm, 0.2971 for Wt <60 μm vs *ΔTRAK2* < 60 μm). **j, k** Plots comparing the velocity of cells versus protrusion length in Wt (**j**) and *ΔTRAK2* (**k**) ECs (*n* = 251 cells Wt, *n* = 310 cells *ΔTRAK2*, two-tailed Pearson's correlation coefficient, ns *P* = 0.8546, \*\*\**P* ≤ 0.0001; *r* = 0.3676). **l** Plots comparing the velocity of Wt or *ΔTRAK2* ECs with protrusions of the indicated length (*n* = 251 cells Wt, *n* = 310 cells *ΔTRAK2*, two-tailed Mann–Whitney test, \*\*\**P* = 0.0007 for 50–70 μm, <0.0001 for 60–80 μm and >70 μm, ns *P* = 0.5453 for 20–40 μm, 0.765 for 30–50 μm, 0.175 for 40–60 μm Wt versus *ΔTRAK2*; two-sided analysis of covariance, magenta asterisk, \*\*\**P* ≤ 0.0001 Wt versus *ΔTRAK2*). Data are mean ± s.e.m. (**c–f, h, i, l**). Source data are provided as a Source Data file.

the control of mitochondria function and cell behaviour in diverse tissue contexts.

## Discussion

Cells exhibit a wide diversity of shapes and undergo broad shifts in morphology during dynamic processes such as cell migration[50]. To match these changes in geometry, cells precisely adjust the size,

number and positioning of organelles to maintain normal physiological function[1–4]. Yet, the mechanisms by which cells sense shifts in their dimensions and tune organelle properties accordingly remain largely unclear. Here, we identify an unexpected role for mRNA transport in the cell-size scaling of organelle behaviour. In particular, we show that trafficking of *TRAK2* mRNA, driven by a single 29 bp 3'UTR motif, fundamentally underpins cell-size-dependent control of mitochondria

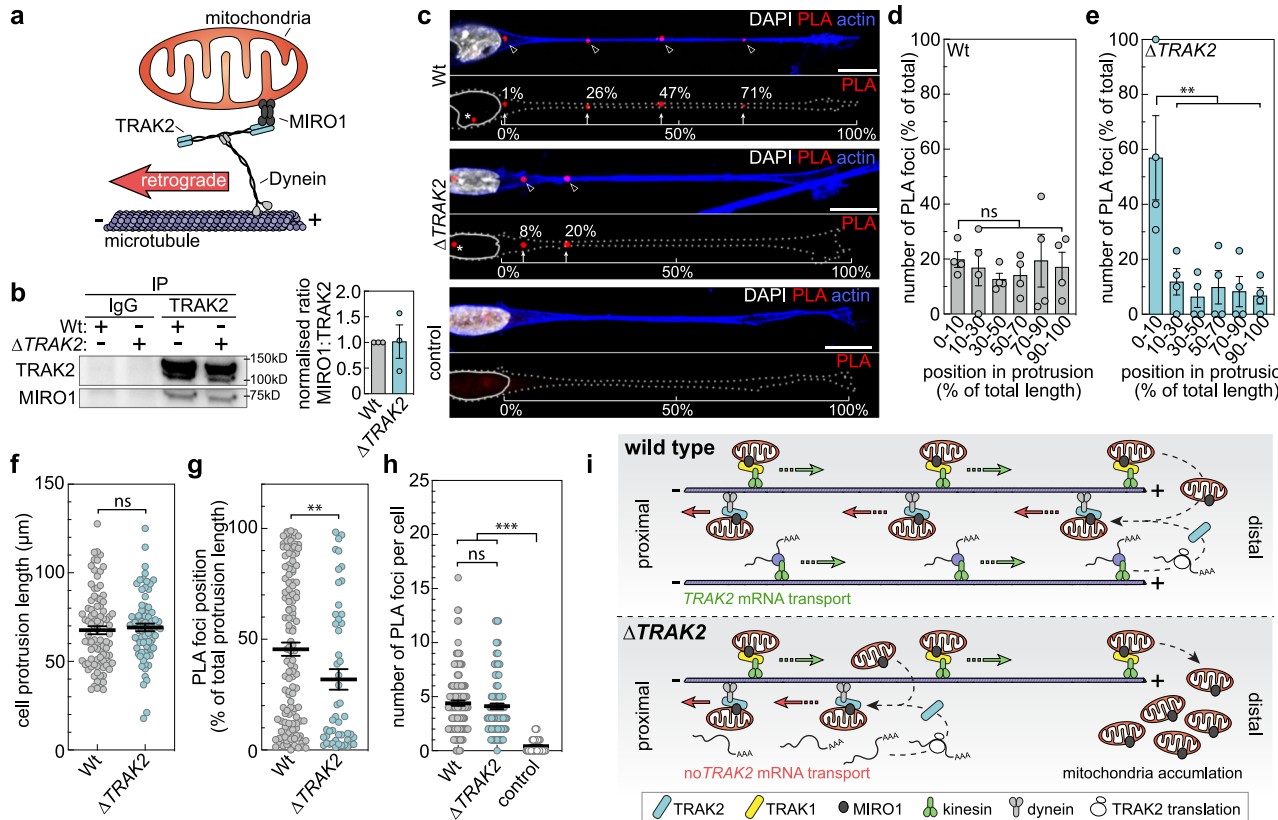

**Fig. 6 | *TRAK2* mRNA localisation spatially modulates TRAK2-MIRO1 interactions. a** Schematic of the interactions between Dynein, TRAK2 and MIRO1 underpinning retrograde transport of mitochondria along microtubules. Figure adapted from Fenton et al.[22] (https://doi.org/10.1038/s41467-021-24862-7) **b** Western blot of TRAK2 and MIRO1 following immunoprecipitation using anti-TRAK2 or control IgG antibodies in Wt or *ΔTRAK2* ECs and densitometric analysis of the ratio of MIRO1:TRAK2 levels (*n* = 3 independent experiments, two-tailed Mann–Whitney test, ns *P* = 0.7000). **c** Proximity ligation assay (PLA) to detect sites of interaction between TRAK2 and MIRO1 in protrusions of Wt or *ΔTRAK2* ECs cultured on CDM. Control PLA was performed in Wt ECs in the absence of primary antibody (arrowheads indicate sites of PLA foci indicative of TRAK2-MIRO1 interaction, brackets and arrows indicate position of PLA foci along protrusions). **d, e** Proportion of PLA foci detected at the indicated positions of protrusions in either Wt (**d**) or *ΔTRAK2* (**e**) ECs (*n* = 4 experiments for Wt, 67 cells total, *n* = 4 experiments for *ΔTRAK2*, 67 cells total, ordinary one-way ANOVA with Tukey's multiple comparisons test, **\*\*P* = 0.0062 for 10–30%, 0.0021 for 30–50%, 0.0042 for 50–70%, 0.0031 for 70–90%, 0.0023 for 90–100% versus 0–10%, ns *P* = 0.9989 for 10–30%, 0.9416 for 30–50%, 0.9771 for 50–70%, >0.9999 for 70–90%, 0.9993 for 90–100% versus

0–10%). **f** Length of protrusions in Wt or *ΔTRAK2* ECs that were used for PLA (*n* = 90 cells Wt, *n* = 77 cells *ΔTRAK2*, two-tailed Mann–Whitney test, ns *P* = 0.3919). **g** Position of individual PLA foci along the length of protrusions in Wt or *ΔTRAK2* ECs (*n* = 124 cells Wt, *n* = 50 cells *ΔTRAK2*, two-tailed Mann–Whitney test, **\*\*P* = 0.0060). **h** Total number of PLA foci in Wt or *ΔTRAK2* ECs. Control PLA was performed in Wt ECs in the absence of primary antibody (*n* = 144 cells Wt, *n* = 116 cells *ΔTRAK2*, *n* = 51 cells control, unpaired Kruskal–Wallis test and Dunn multiple comparison test, ***\*P* ≤ 0.0001 for Wt or *ΔTRAK2* versus control, ns *P* ≥ 0.9999). **i** Schematic of the hypothesised role of *TRAK2* mRNA localisation in the distal interaction of TRAK2 and MIRO1 and modulation of the cellular distribution of mitochondria. It is well-established that kinesin-TRAK1-MIRO1 and dynein-TRAK2-MIRO1 interactions promote anterograde and retrograde transport of mitochondria, respectively[11–22]. We find that *TRAK2* mRNA spatially modulates TRAK2-MIRO1 interactions to support cell-size-scaling of mitochondria distribution. Thus, in elongated cells, mitochondria accumulate at distal sites. Data are mean ± s.e.m. (**b, d–h**). Cells were cultured on CDM. Scale bars, 10 µm. Source data are provided as a Source Data file.

positioning and function. Considering the large number of mRNAs known to be spatially targeted in cells[40–45] and the broad conservation of this phenomenon between diverse cell types[47], it seems unlikely that this phenomenon is restricted to just *TRAK2*. Indeed, the protein product of another targeted mRNA explored in this study, *RASSF3* (Fig. 1d–f, m), was very recently shown to both directly interact with MIRO1 and to impact the subcellular positioning of mitochondria and peroxisomes[54]. Hence, cell-size-dependent spatial targeting of *RASSF3* likely also directs organelle dynamics. Furthermore, both *RAB13* and *KIF1C* mRNAs, which are co-trafficked with *TRAK2* and *RASSF3* (Fig. 1a–l; Supplementary Fig. 3a–f), encode translated protein products that modulate the trafficking of other key organelles, such as endosomes[55,56]. Thus, our work hints that the use of cell-size-dependent mRNA trafficking may be a widespread phenomenon underpinning the size-scaling of diverse organelle dynamics.

Numerous G-motif-containing polarised mRNAs have now been identified[40–45], yet the function of their subcellular trafficking remains

largely unclear. Recent work has elegantly shown that subcellular targeting of some of these mRNAs facilitates interaction of nascent protein with specific binding partners[45,48,49]. Yet here we show that *TRAK2* mRNA trafficking has no impact on the magnitude of TRAK2 association with its key binding partner MIRO1. In contrast, we find that mRNA transport is essential to the spatial control of TRAK-MIRO1 protein complex assembly. Following excision of the 29bp G-motif in *ΔTRAK2* cells, interactions between TRAK2 and MIRO1 are no longer uniformly positioned along the length of the cell but instead predominantly restricted to the proximal zone of protrusions. Thus, *TRAK2* mRNA transport acts to define the spatial domain of protein-protein interactions and the distal-most extent to which mitochondria can be retrieved by TRAK2-MIRO1-mediated retrograde transport. Consequently, excision of a single 29 bp 3'UTR motif in *TRAK2* disrupts homoeostatic control of organelle positioning to trigger progressive distal accumulation of mitochondria as cells increase in size. However, these observations do not entirely rule out an additional role for mRNA

trafficking as a facilitator of interaction with specific TRAK2-binding partners. For example, recruitment of the co-factor Lissencephaly-1 (LIS1) to the TRAK2-dynein-dynactin complex was recently proposed to bias retrograde TRAK2 transport in single-molecule assays using cellular extracts[22]. Considering that LIS1 localises to the leading edge of migrating cells[57], it is indeed feasible that *TRAK2* mRNA transport to distal sites may aid recruitment of LIS1. However, given that TRAK2-MIRO1 still robustly accumulates at proximal sites in *ΔTRAK2* cells, consistent with minus-end-directed transport, any impact of a failure to interact with LIS1 on retrograde transport is likely minimal. Despite these observations, it is still possible that *ΔTRAK2* mutation further impacts mitochondrial transport by spatially modulating interaction with other unidentified TRAK2-binding partners, in addition to MIRO1. As such, future work should fully define (1) the wider spatial interactome of TRAK2, (2) how this is broadly directed by *TRAK2* mRNA localisation, and (3) the impact of any other identified mRNA-mediated protein interactions on mitochondrial motility. Indeed, if other mRNA-regulated TRAK2-binding partner interactions were identified, this may suggest much broader roles for mRNA transport in the control of mitochondrial dynamics. Intriguingly, we find that mRNA encoding another microtubule transport protein, *KIF1C*, is also trafficked to distal sites in a cell-size-dependent manner (Supplementary Fig. 3a–c, g). KIF1C is a member of the kinesin superfamily of proteins that classically promote microtubule plus-end-directed anterograde transport. However, unusual amongst kinesins, KIF1C is also known to direct retrograde microtubule transport[58,59]. Whilst recent work reveals that disruption of *KIF1C* mRNA trafficking perturbs cell migration[49], it remains unclear if this impacts the spatial domain of KIF1C-binding partner interactions, KIF1C-dependent retrograde transport and/or organelle dynamics. As such, it is exciting to speculate a broadly conserved role for trafficking of *KIF1C* and other mRNAs in the spatial control of retrograde trafficking complex assembly and function.

Importantly, here we define a previously unidentified RNA-based mechanism underpinning homoeostatic control of mitochondria distribution and cell motility. Redistribution of mitochondria to the leading edge of migrating cells generates subcellular energy gradients that bias ATP levels at lamellipodia[24–32]. Thus, tight control of mitochondria positioning is a critical determinant of motile cell behaviour[24–32]. Here we find that migrating cells sense shifts in their dimensions and can adjust mitochondria positioning appropriately to maintain a remarkably consistent velocity despite broad changes in cell shape. Moreover, we reveal that *TRAK2* mRNA trafficking fundamentally directs this cell-size-dependent homoeostatic control. Consequently, excision of a single 29 bp *TRAK2* G-motif eradicates cell-size scaling, triggering progressive mitochondria misplacement and hypermotility as cells increase in size. Others have observed that shifts in cell shape have no impact on cell motility[60], and now we provide a potential mechanistic basis for this phenomenon. Moreover, our work identifies that *TRAK2* mRNA dynamics act as a previously unappreciated key control point modulating mitochondria distribution and function. Indeed, previous work has shown that post-translational modification of TRAK2 via glycosylation can dynamically tune mitochondria positioning in response to metabolic change[61]. Regulated shifts in *TRAK2* mRNA positioning and/or abundance would likewise enable rapid modulation of mitochondria spatial distribution. Indeed, our single-molecule analyses reveal surprisingly low *TRAK2* mRNA spot number (~5 per cell), indicating that mitochondrial dynamics would be highly sensitive to even subtle changes in mRNA levels or stability. Yet, whether such shifts in *TRAK2* mRNA dynamics are linked to switches in invasive cell state and/or could be therapeutically manipulated to predictably modulate invasive cell behaviour in disease remains an open question. Likewise, it remains unclear how modulation of *TRAK2* mRNA levels and/or localisation influences other critical aspects of mitochondria dynamics, such as fission, fusion and mitophagy. However, what is clear is that the RNA-binding protein (RBP) machinery

that directs *TRAK2* microtubule plus-end-directed transport must also be a key determinant of cell-size-scaled mRNA transport. Several RBPs are implicated in trafficking of G-motif-containing mRNAs, including APC and UNC[41,44], suggesting that the use of distinct RBPs and/or post-translational modification of RBP function could be employed to fine-tune size-scaled responses in distinct tissue contexts. Taken together, this work sheds light on a previously unappreciated role for mRNA transport in the cell-size-dependent modulation of organelle dynamics that has wide implications for understanding the control of mitochondrial function and motile cell behaviour in health and disease.

## Methods

### Cell culture, transfections, and cell-derived matrix production

All cells were maintained at 37 °C, 5% $CO_2$, with regular passages. HUVEC (PromoCell, Cat. No. C-12200) and hCMEC/d3 (Poller et al., 2008; Merc, Cat.No. SCC066) were cultured on 0.1% gelatin (Millipore) coated T75 flasks (Corning) in endothelial cell basal medium 2 (EBM-2, Promocell) with an additional supplement pack (5% FCS, EGF (5 ng ml⁻¹), VEGF (0.5 ng ml⁻¹), FGF2 (10 ng ml⁻¹), long R3 insulin growth factor-1 (20 ng ml⁻¹), hydrocortisone (0.2 μg ml⁻¹), and ascorbic acid (1 μg ml⁻¹)). 50 mg ml⁻¹ gentamycin (Sigma) and 250 μg ml⁻¹ amphotericin (Sigma) were also added to the culture media. All HUVEC experiments were conducted with cells between passages 3–6. Mouse Balb/c brain endothelioma cells (b.End5, ATCC; Merc, Cat. No. 96091930) and Telomerase-Immortalised Fibroblasts (TIFs; Gift from Prof. P.T. Caswell, University of Manchester) were cultured in Dulbecco's Modified Eagle Medium (DMEM, Sigma) supplemented with 10% FCS and 10 U ml⁻¹ Penicillin-Streptomycin. All cells were detached using Trypsin-EDTA (Sigma) and were split before reaching 90% confluency, with media being changed every 2 days.

Gene knockdown was achieved using ON-TARGETplus non-targeting or gene-specific SMARTpool siRNAs (Horizon). 0.3 μM siRNA was diluted in 200 μl Opti-MEM containing 1.5% v/v GeneFECTOR (Venn Nova). This mix was then added to a well of a sixwell plate containing 1 ml Opti-MEM and the sample was incubated at 37 °C for 3 h. Opti-MEM was then replaced with endothelial growth media and samples were allowed to recover for 48–72 h.

Cell-derived matrix (CDM) was produced according to previously described protocols (Cukierman et al., 2001; Franco-Barraza et al., 2016). Briefly, 0.2% gelatin-coated (Sigma) cell culture dishes or coverslips were fixed with 1% Glutaraldehyde (Sigma), before quenching with 1 M Glycine (Sigma) and equilibration with DMEM. TIFs were seeded at full confluency and grown for 8 days with 25 μg ml⁻¹ Ascorbic Acid (Sigma), changing the media every 2–3 days. TIFs were then denuded with 20 mM NH4OH, 0.5% Triton-X-100 in PBS. Finally, samples were treated with 10 μg ml⁻¹ DNAse I (Roche) before seeding of endothelial cells.

For MS2 transfections, 1×10⁵ b.End5 cells were cultured in 35 mm round glass-bottom dishes (Greiner CellView) and transfected with the following: 0.5 μg pcDNA3.1-Lyn-Cherry, 1 μg pCS2-MCP-GFPnls, 1 μg different versions of pcDNA3.1-HBB-24xMS2SL-3′UTR. Lipofectamine 3000 (Thermo) was used following manufacturer's instructions. Cells were incubated for 48 hours before microscopic analysis.

### Generation of CRISPR-Cas9 mutant cell lines

crRNAs were designed to target sequences immediately surrounding the minimal localisation elements in the *TRAK2* 3′UTRs using the online IDT Alt-R CRISPR-Cas9 design tool to minimise off-target effects (Supplementary table 1). IDT Alt-R CRISPR-Cas9 protocol was followed mostly according to manufacturer's instructions. Briefly, 200 μM each crRNA was hybridised to 200 μM tracrRNA by heating to 95 °C in a thermal cycler and allowing to cool to room temperature. RNP complexes were formed by mixing 120 pmol crRNA:tracrRNA duplex with 104 pmol Alt-R Cas9 in PBS and incubating at room temperature for

20 min. RNP complexes for generating wt cell lines contain no crRNA:tracrRNA duplexes.

RNP complexes were added to $5 \times 10^5$ hCMEC/d3 cells along with 2 µg pMAX-GFP plasmid (Lonza) and transfection was carried out using a Nucleofector 2b (Lonza) according to the manufacturer's instructions. Cells were then cultured for 72 hours before individual GFP-expressing clones were sorted into wells of a 96-well plate using a FACSAria Fusion cell sorter (BD Biosciences). Clonal populations were screened for genomic lesions using gene-specific primers to amplify regions in the 3′UTRs by PCR (Supplementary Table 1). Two Wt and four *ΔTRAK2* lines were created and used in combination for each experiment.

## smFISH and fluorescence labelling

Endothelial cells on glass coverslips or CDM were fixed in methanol-free 4% formaldehyde and permeabilised in 70% ethanol. Stellaris single-molecule fluorescence in situ hybridisation (smFISH) gene-specific probe sets (Supplementary Table 2), conjugated to Quasar 570/670 fluorophores, were designed using an online tool (Biosearchtech). Predesigned GAPDH reference probe sets are purchased directly. smFISH probe hybridisation was carried out as described previously[40]. Briefly, samples were washed in wash buffer (2× SSC, 10% formamide) before incubation overnight at 37 °C with smFISH probe sets diluted in hybridisation buffer (2× SSC, 10% formamide, 10% w/v Dextran Sulphate). Samples were then washed twice in wash buffer at 37 °C for 30 min. To co-label nuclei and F-actin, samples were further incubated with 5ug ml$^{-1}$ DAPI (Sigma) in PBS, and with 165 nM Alexa-Fluor 488/647 Phalloidin in PBS (Thermo Fisher Scientific), for 5 mins at room temperature. To label mitochondria, MitoTracker was added to the cell media at 250 nM for 30 min. Fresh media were then added, and cells were incubated for 2 hrs before fixation. To label microtubules, cells were fixed in 4% PFA at room temperature for 10 min, then permeabilised in ice-cold methanol at 4 °C for 20 min. Cells were then stained with anti-alpha tubulin antibody (Abcam ab7291) 1:100, followed by a goat anti-mouse AlexaFluor 568-conjugated secondary antibody (1:500, Thermo Fisher Scientific). All fixed samples were mounted using Prolong Gold (Thermo Fisher Scientific). For TMRM staining, cells were seeded onto glass-bottom dishes in EBM-2 1–2 days before imaging.

Growth media was replaced with fresh media containing 200 nM Mitotracker Green FM 9074 (Cell Signalling Technology) and 1× Image-iT TMRM Reagent (Invitrogen). Cells were incubated for 30 min and then washed 1× with PBS before fresh media was added prior to imaging.

## Plasmid construction

All plasmids for in vitro MS2 experiments were generated as described previously[40]. Briefly, 3′UTR sequences were amplified by PCR using sequence-specific primers (Supplementary table 1) from human genomic DNA using MyTaq Red Mix (Bioline) according to manufacturer's instructions. Site-directed mutagenesis was used to create precise deletions in 3′UTRs. Primers for SDM were designed using Agilent QuickChange II design software. Phusion (NEB) was then used with the following reaction conditions: denaturation 95 °C for 30 s, annealing 55 °C for 1 m 30 s, extension 72 °C for 6 m for 18 cycles. Reactions were then treated with DpnI (NEB) for a minimum of 1 hr at 37 °C. 3′UTR sequences were then cloned into a pcDNA3.1-HBB-24xMS2SL-MCS expression vector using the NheI and XhoI/ApaI restriction sites in the multiple cloning site[40].

## Microscopy

Unless otherwise stated, fluorescent microscopy data were obtained using an Olympus IX83 inverted microscope using Lumencor LED excitation, a ×100/1.40 UplanSApo objective and a Sedat QUAD filter set (Chroma 89000). The images were collected using an R6

(Q-imaging) CCD camera with a Z optical spacing of 0.2 µm. Raw images were then deconvolved using the Huygens Pro software (SVI), and maximum intensity projections of these deconvolved images are shown in the results. To image microtubules, cell were imaged using a Zeiss LSM 980 Airyscan microscope using a Zeiss Plan-APOCHROMAT ×63/1.4 Oil Ph3 objective and a Z-stack spacing of 0.5 µm. For TMRM staining, images were acquired on an Eclipse Ti inverted microscope (Nikon) using a ×20/0.45 SPlan Fluor or ×40/0.6 SPlan Fluor objective, Nikon filter sets for brightfield, GFP and mCherry and a pE-300 LED (CoolLED) fluorescent light source. Throughout imaging, cells were maintained at 37 °C and 5% CO$_2$. Images were then collected using a Retiga R6 (Q-Imaging) camera with a Z optical spacing of 0.5 µm. Cell motility tracking data was obtained using an Eclipse Ti inverted microscope (Nikon) using a ×10/0.45 Plan Fluor (Ph1DLL) objective, and a pE-300 LED (CoolLED) fluorescent light source. Imaging software NIS Elements AR.46.00.0. Point visiting was used in combination with laser-base autofocus to allow multiple positions to be imaged within the same time-course, and cells were maintained at 37 °C and 5% CO$_2$. The images were collected using a Retiga R6 (Q-Imaging) camera.

## Immunoprecipitation

ECs were trypsinized, pelleted, washed twice in PBS, pelleted and then lysates prepared upon resuspension in 500 µl ice-cold NP-40 lysis buffer (20 mM Tris-HCl pH 8, 137 mM NaCl, 10% glycerol, 1% Nonidet P-40, 2 mM EDTA, 1:100 protease inhibitor cocktail). Lysates were rotated for 1 hr at 4 °C, centrifuged at 13,500 × *g* for 30 min and supernatant collected. 10 µg of anti-TRAK2 antibody or 10 µg control IgG antibody was then added to EC lysates and incubated at 4 °C overnight with mixing. 0.25 mg Protein A/G Magnetic Beads (Pierce) were washed in NP-40 lysis buffer, then the antibody/lysate mixture was added and incubated at room temperature for 1 hr with mixing. Beads were then collected, washed twice in NP-40 lysis buffer and then once in purified water. Beads were finally collected, incubated with 40 µl of 1× Laemmli buffer, mixed on a rotor for 15 min at room temperature and then supernatant collected in preparation for western blotting.

## Western blotting

Western blotting was performed using either whole cell protein samples or immunoprecipitation supernatant. Whole cell protein was extracted using RIPA buffer (25 mM Tris−HCl pH 7.6, 150 mM NaCl, 1% NP-40, 1% sodium deoxycholate and 0.1% SDS) containing 1:100 Protease Inhibitor Cocktail. Protein concentration was quantified using Pierce BCA Protein Assay Kit (Thermo Fisher Scientific) before denaturation with Laemmli Buffer (250 mM Tris−HCl pH 6.8, 2% SDS, 10% glycerol, 0.0025% bromophenol blue, 2.5% β-mercaptoethanol) at 95 °C for 5 min. Proteins were separated using 10% Mini-PROTEAN TGX precast protein gels (Bio-Rad). Trans-Blot Turbo transfer system (Bio-Rad) was used to transfer proteins to PVDF membranes using manufacturer's guidelines. Membranes were then blocked for 1 hr at room temperature in 2.5% BSA (Sigma) in TBS containing 0.1% Tween-20. Primary antibodies were diluted in blocking buffer and incubated overnight at 4 °C. Membranes were then washed with TBS containing 0.1% Tween-20 before incubation with secondary antibodies diluted in blocking buffer for 1 hr at room temperature, followed by more washes. Pierce SuperSignal West Atto Western blotting substrate (Thermo Fisher Scientific) was used to develop chemiluminescent signal, which was then detected digitally using a ChemiDoc MP Imager (Bio-Rad).

## Antibodies

Primary and secondary antibodies were used at the following concentrations: rabbit anti-TRAK2 (1:1000 Western blot, Proteintech, 13770-1), mouse anti-TRAK2 (1:500 Western blot, Thermo Fisher Scientific, MA5-27606), mouse anti-MIRO1 (1:200 Western blot, proximity

ligation assay, Abcam, CL1083), rabbit anti-β-actin (1:1000 Western blot, Cell Signalling technology, 4967), rabbit polyclonal anti-TRAK2 (1:50 Immunoprecipitation, 1:200 proximity ligation assay, Thermo Fisher Scientific, PA5-34889), mouse anti-alpha tubulin (1:100 Abcam ab7291), rabbit IgG (1:50 Immunoprecipitation, Proteintech, 30000-0-AP), goat anti-mouse AlexaFluor 568 (1:500 Immunofluorescence, Thermo Fisher Scientific, A-21043), goat anti-rabbit AlexaFluor 488 (1:500 Immunofluorescence, Thermo Fisher Scientific, A-11008).

## Image analysis

RNA spot counts were obtained using Fuji (2.14.0/1.54j) plugin FindFoci[62]. This tool uses an automated Otsu thresholding algorithm for detecting peaks with intensity above the background. RNA spots with an intensity peak above the threshold were then counted and highlighted with red circles in images. RNA polarisation measurement was obtained for background-subtracted smFISH images using the previously described Polarisation and Dispersion Index[53]. The Polarisation Index assesses the extent to which an mRNA is either centrally localised or peripherally polarised within a defined cell shape. Here, if an mRNA is polarised, the centroid of the mRNA should be distinct from the centroid of the cell. The Polarisation Index thus defines the displacement vector pointing from the cell centroid to the mRNA centroid. This polarisation vector is then divided by the radius of gyration of the cell, which is calculated by the root-mean-square distance of all pixels within an image of a cell from the cell centroid. Hence, the Polarisation Index defines the polarisation of the mRNA normalised to the size and the elongation of a cell. The Dispersion Index determines the extent to which an mRNA is either clustered together or uniformly distributed within a defined cell shape. Here, mRNA dispersion is quantified by determining the second moment $\mu 2$ of RNA positions, which is dictated by the shape and size of the cell and mRNA distribution. Within a cell of defined shape, the Dispersion Index calculates the second moment $\mu 2$ for both the test mRNA and a hypothetical mRNA of uniform distribution. Consequently, dividing the test mRNA second moment $\mu 2$ by the second moment $\mu 2$ of the hypothetical uniform mRNA thus normalises the test mRNA dispersion to cell shape. Alternatively, RNA polarisation was also measured by calculating the XY coordinates of the centre of mass (CoM) of the RNA fluorescence (using the Analyse menu in Fuji (2.14.0/1.54j)) and determining the Euclidean distance of the RNA CoM from the nearest edge of the nucleus. For some analyses, the distance that the RNA centre of mass (CoM) sits from the EC nucleus was also normalised to protrusion length. For quantification of the fluorescence intensity of labelled mitochondria, sum intensity projections of raw image files were generated in Fuji (2.14.0/1.54j), cell outlines and other regions of interest (ROI) were defined using the polygon tool and integrated density of fluorescence of ROIs and background measured, along with the ROI area. Cell motility tracking in CDM was performed manually using the MTrackJ plugin for Fuji (2.14.0/1.54j). Randomly selected cells were tracked through each frame of the time-lapse movies. Quantification of cells on glass was achieved using the Chemotaxis and Migration Tool (Ibidi). For cells both in CDM and on glass, directionality was calculated as the displacement of a cell from point A to B, divided by the total length of the path taken to get there. EC morphometrics (aspect ratio, circularity and protrusion length) were quantified using Fuji (2.14.0/1.54j). For analysis of motile cell protrusions, firstly, these protrusions were identified as the cell protrusion exhibiting either the highest enrichment of localised mRNAs, polarised perinuclear enrichment of mitochondria or enrichment of TRAK2-MIRO1 PLA foci. As cells were predominantly uniaxially elongated, motile protrusions were usually easily distinguishable as the longest protrusion extended by a cell. To quantify protrusion length, the distance from the nuclear envelope to the leading edge of the cell was then determined.

## Proximity ligation assays

ECs cultured on CDM in glass-bottomed dishes were first fixed in 4% PFA for 10 min, washed twice for 5 min in PBS and then permeabilised in 0.1% PBS Triton-X for 30 min at room temperature. Proximity ligation assays were then performed using the Duolink PLA fluorescence kit (Merck), as per manufacturer's guidelines. Of note, the volume of incubations was set at 120 µl per glass-bottomed dish, and primary antibodies were incubated overnight at 4 °C. Moreover, prior to mounting, samples were incubated with Phalloidin-Alexafluor 488 (1:40 dilution) for 20 min.

## Statistics & reproducibility

All statistical analyses were performed using GraphPad Prism 10 (v.10.23(347)) software. No statistical methods were used to predetermine sample sizes. Normality testing (Shapiro–Wilk) was performed to determine if datasets were suitable for parametric or non-parametric statistical tests. All data are presented as mean ± s.e.m., unless stated otherwise. Either a two-tailed unpaired Student's t-test or a two-tailed Mann–Whitney test was used for the comparison of two normally distributed or non-normally distributed datasets, respectively. An ordinary one-way ANOVA with Tukey's multiple comparisons test was used to compare more than two normally distributed datasets. To compare more than two non-normally distributed datasets, an unpaired Kruskal–Wallis test and Dunn multiple comparison test were used. Data correlation was tested via the Pearson's correlation coefficient. Differences between linear regressions were assessed via a two-sided analysis of covariance. Error bars, P values, n numbers and statistical tests used are reported in the figures and/or figure legends. All statistical tests were performed on datasets acquired from at least two independent experiments. For each analysis, measurements were taken from distinct samples.

## Reporting summary

Further information on research design is available in the Nature Portfolio Reporting Summary linked to this article.

## Data availability

All data generated or analysed during this study are included in this published article (and its supplementary information files). Source data are provided with this paper.

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

## Acknowledgements
We thank the University of Manchester Bioimaging Core, Biological Services and Flow Cytometry Facilities for technical support. We also thank members of the S.P. Herbert and R. Das laboratories for discussion. This work was supported by a fellowship grant from the Wellcome Trust (219500/Z/19/Z) to S.P.H. and a BBSRC DTP PhD studentship to J.J.B.

## Author contributions
S.P.H. conceived and designed the project. J.J.B. performed experimental studies with assistance from G.E.H., R.V., G.C. and H.E.L. S.P.H. supervised the study. The manuscript was written by S.P.H. and J.J.B.

## Competing interests
Authors declare no competing interests.
