## [Transparent Peer Review file · Nature Communications]

mRNA trafficking directs cell-size-scaling of mitochondria distribution and function

Corresponding Author: Dr Shane Herbert

Version 0:

Reviewer comments:

Reviewer #1

(Remarks to the Author)

Mitochondrial trafficking within cells is primarily driven by motor proteins such as kinesin-1 and dynein and is mediated by motor adaptor proteins like TRAK1 and TRAK2, which connect motors to the mitochondria through the outer mitochondrial membrane protein Miro. In this paper Bradbury et al. show that the precise localization of TRAK2 mRNA plays a role in regulating mitochondria distribution in the cell and maintaining its cell-size scaling. The authors identify a conserved 3'UTR GA-rich motif of TRAK2 mRNA which is responsible of targeting of TRAK2 to distal sites in cell protrusions and determines for retrograde transport of mitochondria.

Overall, this is a solid and highly interesting study providing significant insights into the role of TRAK2 mRNA in mitochondrial dynamics. The paper is well-written, the experiments are solid and conclusive, and the figures are clear and of good quality. I am left with just a couple of questions:

1. The authors state that enrichment of transcripts at the tip of protrusions is more acute in cells on CDM versus glass and they refer to Figures 1 b,e,h. However, in these figures presenting the distance of mRNA (CoM) from the nucleus, distance is not normalized. The presented increased distance does not necessarily suggest that the enrichment at the tips is more pronounced but could simply mean that the cells are longer. To support their original statement the authors should normalize the values to the length of the cells.
2. Figure 2, can the authors confirm that knock down of TRAK2 does not influence the microtubule cytoskeleton in protrusions?
3. Can the authors discuss if the deletion in Δ TRAK2 could affect interactions of TRAK2 with its interaction partners and consequently affect motility of mitochondria.
4. RNA spots in figures such as 3c seem to have different sizes (fluorescence intensities). Was the size/intensity reflected in data analysis to obtain RNA spots counts or frequencies? Was the image thresholded? Please give more details on these methods. For example, in Figure 4e on the x axis is it really number of mRNA or is it rather fluorescence signal?
5. Can the authors clarify how did they calculate Polarization and Dispersion index and what do these parameters mean.
6. Can the authors explain what they mean with "distinct scaling properties"(page 6, line 25)
7. How is the directionality in figure 5 e defined and calculated.
8. Page 16 line 20 - this sentence suggests that kif2C drives the retrograde transport, this would be probably good to rephrase.
9. Introduction is slightly convoluted with results. Please clarify what data is from previous experiment and what data is new.
 - Differential enrichment in cell protrusions and cell bodies (extended data Fig1.)
 - Fimo analysis – please explain for nonspecialist readership, what is FIMO analysis. Why is data not shown?

- Some sentences in Figure 1 would fit better in Introduction.

Reviewer #2

(Remarks to the Author)

In this manuscript titled 'mRNA trafficking directs cell-size-scaling of mitochondria distribution and function', the authors set out to understand how mitochondrial positioning changes in a cell that undergoes dynamic shape changes. They characterize the localization of mRNA containing GA-rich motifs, specifically that of TRAK2 to show that: (i) TRAK2 mRNA is enriched towards the protrusion tip in endothelial cells moving on glass substrates and even more so on cell-derived extracellular matrix (CDM), (ii) mitochondria are enriched at the protrusions of cells that are depleted of TRAK2, (iii) a 29-bp GA-rich motif in TRAK2 removes the protrusion-dependent scaling of TRAK2 mRNA and phenocopies the TRAK2 RNAi mitochondrial localization (iv) cells that have the GA-rich motif knockout have altered speeds of movement on CDM (but not glass) (v) localized translation of TRAK2 mRNA and association of the resulting TRAK2 with MIRO1 is likely the mechanism by which mitochondrial position is maintained as the cell changes shape.

While the underlying mechanism of cell-size-dependent mitochondrial scaling is an important premise, this manuscript fails to show conclusively that TRAK2 mRNA enrichment at the cell protrusions underpins mitochondrial positioning or function. I therefore cannot recommend the article to be published in its current form. My specific comments are:

1. The title alludes to mitochondrial function - however, there were no experiments conducted to test the functional status of mitochondria. This would be essential to claim that mitochondrial function is changed in cells without TRAK2 mRNA enrichment at cell protrusions.
2. While the authors ruled out differences in protrusion shape/size as factors underlying accumulation of mitochondria at protrusions, the authors should also ensure that the underlying microtubule network is unchanged in control and TRAK2 siRNA/DelTRAK2 cells.
3. It is unclear why mitochondrial accumulation at the protrusion tip would lead to increased velocity and movement of cells on CDM. The authors refer to previous work where mitochondria appear polarized to the leading edge in migrating cells, but the mitochondria in DelTRAK2/siRNA cells does not appear to be polarized. This part of the work needs to be developed extensively to make any claims about mitochondrial function being altered and thereby leading to increased motility of cells.
4. So too, the role of motor proteins esp. cytoplasmic dynein in mediating the proposed TRAK2-MIRO1 facilitated retrograde mitochondrial transport has not been shown and is key to the conclusions of the paper.
5. The number of PLA spots showing interaction between TRAK2 and MIRO1 (by the authors' own admission) is only 4-5, even in WT controls. Additionally, MIRO1(/2)-independent TRAK1/2-dependent mitochondrial transport has been established in MEFs (DOI: 10.15252/emboj.201696380) - how important is the (loss of) TRAK2-MIRO1 interaction?
6. Fig. 2: The mitochondrial intensities in TRAK2 RNAi and delTRAK2 have been normalized to the corresponding segments of the protrusions in control cells. However the total mitochondrial intensity seems altered in TRAK2 RNAi/DelTRAK2- it would be ideal to quantify mitochondrial intensity as a proportion of the total mitochondrial intensity *within* each cell so as to show true enrichment. So too, quantifying the mitochondrial intensity per unit volume of each segment is a better measure.
7. The only part of the schematic that appears in Fig. 6i that has been demonstrated is the enrichment of TRAK2 mRNA at protrusions and perhaps TRAK2-MIRO1 interactions. The rest of the mechanism is speculative, and has to be demonstrated - via live cell imaging, photoactivation/photoconversion experiments of the mitochondria and by visualising/disrupting the motors in control and TRAK2 siRNA/DelTRAK2 cells.
8. The data from independent experiments seem to have been pooled and the stats seem to have been performed on the number of cells. It would be good to indicate the different replicates (using diff colors) and to do the stats on the independent repeats (n=3 for instance, rather than n=no. of cells)
9. The smFISH images need to be represented accurately - the contrast in these images seems to have been set at a level where only individual punctae are visible, but it would be essential to provide raw images where the background can also be seen. So too, some smFISH images which contain the red mRNA spots are not accompanied by the original images - this makes it very hard to actually see where the spots are.
10. Overall the methods need to be better described - how is a protrusion identified/defined? Where is the protrusion length measured from? The centroid of the nucleus? Each of the figures need to have extensive methods sections.
11. Some terms (e.g.: "FIMO analysis") have been introduced without context which make parts of the manuscript hard to follow. So too, having the phrase "(data not shown)" is not ideal - surely there must be a way to show the data?

Reviewer #3

(Remarks to the Author)

Version 1:

Reviewer comments:

Reviewer #1

(Remarks to the Author)

All my comments (which were anyway minor) have been addressed by the authors and I am in favor of publication of the manuscript in its current form.

Reviewer #2

(Remarks to the Author)

The authors have sufficiently addressed my questions and comments.

Reviewer #3

(Remarks to the Author)

REVIEWER COMMENTS

Reviewer #1 (Remarks to the Author):

Mitochondrial trafficking within cells is primarily driven by motor proteins such as kinesin-1 and dynein and is mediated by motor adaptor proteins like TRAK1 and TRAK2, which connect motors to the mitochondria through the outer mitochondrial membrane protein Miro. In this paper Bradbury et al. show that the precise localization of TRAK2 mRNA plays a role in regulating mitochondria distribution in the cell and maintaining its cell-size scaling. The authors identify a conserved 3'UTR GA-rich motif of TRAK2 mRNA which is responsible of targeting of TRAK2 to distal sites in cell protrusions and determines for retrograde transport of mitochondria.

Overall, this is a solid and highly interesting study providing significant insights into the role of TRAK2 mRNA in mitochondrial dynamics. The paper is well-written, the experiments are solid and conclusive, and the figures are clear and of good quality. I am left with just a couple of questions:

1. The authors state that enrichment of transcripts at the tip of protrusions is more acute in cells on CDM versus glass and they refer to Figures 1 b,e,h. However, in these figures presenting the distance of mRNA (CoM) from the nucleus, distance is not normalized. The presented increased distance does not necessarily suggest that the enrichment at the tips is more pronounced but could simply mean that the cells are longer. To support their original statement the authors should normalize the values to the length of the cells.

Author comment: The reviewer correctly points out that, to claim in Fig.1b,e,h that mRNAs are more 'enriched' at distal sites in cells on CDM versus glass substrates, the position of mRNAs should be normalised to cell length. Indeed, we have already performed this important normalisation of mRNA positioning to cell length, however, this data is not presented/discussed until much later in the figure (Fig.1m). To aid clarity, when discussing the earlier observations of mRNA positioning on CDM versus glass substrates, which lack normalisation, we have now replaced the term "enrichment" with the more appropriate term, "displacement", (page 6 line 12).

2. Figure 2, can the authors confirm that knock down of TRAK2 does not influence the microtubule cytoskeleton in protrusions?

Author comment: As requested, we now include new data confirming that the microtubule network is indeed unperturbed upon knockdown of TRAK2 (new Supplementary Fig.4a,b; page 7, lines 23-24).

3. Can the authors discuss if the deletion in Δ TRAK2 could affect interactions of TRAK2 with its interaction partners and consequently affect motility of mitochondria.

Author comment: The reviewer raises a key point that warrants further discussion in the text. Firstly, the deletion in Δ TRAK2 is a 29bp excision in the non-coding 3'UTR, which does not impact the coding sequence or production of full-length protein (see Supplementary Fig.6g). As such, the Δ TRAK2 deletion only impacts mRNA localisation and itself should not render TRAK2 protein unable to interact with its binding partners. Indeed, the Δ TRAK2 mutation does not impact the 'magnitude' of TRAK2 interaction with MIRO1, but instead 'spatially'

defines the site of this key interaction due to shifts in mRNA localisation. The resulting loss of TRAK2-MIRO1 interactions at distal sites thus can largely explain why mitochondria accumulate at distal regions (as this interaction is critical to minus-end-directed mitochondria transport). Yet, we acknowledge that this does not rule out that perturbation of *TRAK2* mRNA trafficking could also impact interaction with other TRAK2-binding partners that influence mitochondria motility (for example LIS1, as detailed in the discussion). Despite this, any perturbation to minus-end-directed TRAK2-MIRO1-mitochondria transport is likely minimal, considering that TRAK2-MIRO1 still robustly accumulates at proximal sites (see full discussion on page 16 lines 14-22). Indeed, if perturbation of minus-end-directed transport did underpin the observed distal accumulation of mitochondria, TRAK2-MIRO1 might be expected to accumulate at distal sites – which is not the case. However, as implied by the reviewer, further discussion should be added to emphasise that future work is needed to fully define (1) the wider ‘spatial interactome’ of TRAK2, (2) how this is broadly directed by *TRAK2* mRNA localisation, and (3) the impact of any other mRNA-mediated protein interactions on mitochondria motility. Indeed, if other mRNA-regulated TRAK2-binding partner interactions were identified, this would suggest much broader roles for mRNA transport in the control of mitochondria dynamics. We now include discussion of this key point on page 16, lines 22-25, and page 17, lines 1-2.

4. RNA spots in figures such as 3c seem to have different sizes (fluorescence intensities). Was the size/intensity reflected in data analysis to obtain RNA spots counts or frequencies? Was the image thresholded? Please give more details on these methods. For example, in Figure 4e on the x axis is it really number of mRNA or is it rather fluorescence signal?

Author comment: We apologise for the lack of detail regarding the quantification of RNA spots. Firstly, some confusion likely arises from occasional inaccurate use of the term “number of mRNAs” when discussing the number of RNA spots. Both the text and figures have been edited to correct this issue. Moreover, to clarify the method used, RNA spots were counted using the well-established FindFoci ImageJ plugin (doi: 10.1371/journal.pone.0114749). This tool uses an automated Otsu thresholding algorithm (doi: 10.1109/TSMC.1979.4310076) for detecting peaks with intensity above the background. Thus, whilst spot size is not directly considered, confirmed RNA spots highlighted with red circles in the images are those with an intensity peak above threshold, as identified by FindFoci. These details are now included in the Methods section (Page 24, lines 19-22).

5. Can the authors clarify how did they calculate Polarization and Dispersion index and what do these parameters mean.

Author comment: Again, we apologise for omitting details of the calculation and utility of these indices. These details have now been added to the Methods section (page 24, lines 23-26, and page 25 lines 1-12). To clarify, both the Polarisation Index and Dispersion Index are routinely applied to smFISH datasets to enable an unbiased description of shifts in localisation of mRNAs (doi.org/10.1016/j.celrep.2011.12.009).

Firstly, the Polarisation Index assesses the extent to which an mRNA is either centrally localised or peripherally polarised within a defined cell shape. Here, if an mRNA is polarised, the centroid of the mRNA should be distinct to the centroid of the cell. The Polarisation Index thus defines the displacement vector pointing from the cell centroid to the mRNA centroid. This polarization vector is then divided by the radius of gyration of the cell, which is calculated by the root-mean-square distance of all pixels within an image of a cell

from the cell centroid. Hence, the Polarisation Index defines the polarisation of the mRNA normalized to the size and the elongation of a cell.

Secondly, the Dispersion Index determines the extent to which an mRNA is either clustered together or uniformly distributed within a defined cell shape. Here, mRNA dispersion is quantified by determining the second moment μ^2 of RNA positions, which is dictated by the shape and size of the cell, and mRNA distribution. Within a cell of defined shape, the Dispersion Index calculates the second moment μ^2 for both the test mRNA and a hypothetical mRNA of uniform distribution. Consequently, dividing the test mRNA second moment μ^2 by the second moment μ^2 of the hypothetical uniform mRNA thus normalises the test mRNA dispersion to cell shape. A uniformly distributed mRNA will have a Dispersion Index of 1. As an mRNA becomes increasingly clustered, it will have a Dispersion Index that is increasingly less than 1.

6. Can the authors explain what they mean with “distinct scaling properties”(page 6, line 25)

Author comment: As cells increase in length, we observed that distinct mRNAs respond very differently to this increase in size scale. In some cases, mRNA enrichment at distal sites is significantly enhanced as cells increase length (e.g. *RASSF3* and *RAB13*; see Fig.1m). In contrast, for *TRAK2*, mRNA localisation is consistently maintained at a position ~60% along the length of cell protrusions as cells increase size (see Fig.1m). As such, whereas the RNA centre of mass for *RASSF3* and *RAB13* are sensitive to shifts in cell size, the precise subcellular distribution of *TRAK2* mRNA is highly robust to changes in size scale. Consequently, we refer to the localisation of these different mRNAs as exhibiting “distinct scaling properties”. To further clarify this point, we have added additional text to page 7, lines 3-5.

7. How is the directionality in figure 5 e defined and calculated.

Author comment: We apologise for not including this information in the Methods section of the original submission. Directionality is a well-established measure that is both defined and calculated as the displacement of a cell from point A to B, divided by the total length of the path taken to get there. As cell migration becomes more directional and less random, then the closer the quantified cell directionally will be to a value of 1. We have now added this information to the Methods (page 25, lines 23-24).

8. Page 16 line 20 - this sentence suggests that kif2C drives the retrograde transport, this would be probably good to rephrase.

Author comment: We apologise for any confusion, as we could not find any mention of KIF2C in the manuscript. However, if this point refers to KIF1C, then the indicated text refers to the unusual non-canonical role of KIF1C in indeed promoting dynein-dependent retrograde transport (see doi.org/10.1038/s42003-024-07023-6; doi.org/10.1038/s41467-019-10644-9; doi.org/10.1083/jcb.201812170). As such, to the best of our knowledge, this statement is correct.

9. Introduction is slightly convoluted with results. Please clarify what data is from previous experiment and what data is new.

- Differential enrichment in cell protrusions and cell bodies (extended data Fig1.)

Author comment: We apologise for the lack of clarity and have added new text (page 4, lines 12-13) detailing that these observations were made upon re-analysis of our previously published dataset defining protrusion-enriched RNAs ([doi.org:10.15252/embj.2020106003](https://doi.org/10.15252/embj.2020106003)).

- Fimo analysis – please explain for nonspecialist readership, what is FIMO analysis. Why is data not shown?

Author comment: We have added new text confirming that the ‘Find Individual Motif Analysis’ (FIMO) program is a search tool for identifying known RNA/DNA motifs in any sequence of interest (Page 4, lines 18-21). Moreover, we have removed reference to “data not shown”, as this refers to a negative result from the FIMO analysis, confirming that the GA-rich motif is not detected in *TRAK1* mRNA. As such, there is no data output to show, and a mention of the lack of a detectable GA-rich motif is sufficient.

- Some sentences in Figure 1 would fit better in Introduction.

Author comment: We thank the reviewer for this comment and agree that text in the Fig.1 results section describing how motile cells exhibit dynamic shape changes and thus must reposition organelles accordingly would better fit in the introduction. As such, we have moved this text and restructured the start of the Fig.1 results section accordingly (page 5, lines 4-15, and page 6, lines 4-8).

Reviewer #2 (Remarks to the Author):

In this manuscript titled ‘mRNA trafficking directs cell-size-scaling of mitochondria distribution and function’, the authors set out to understand how mitochondrial positioning changes in a cell that undergoes dynamic shape changes. They characterize the localization of mRNA containing GA-rich motifs, specifically that of TRAK2 to show that: (i) TRAK2 mRNA is enriched towards the protrusion tip in endothelial cells moving on glass substrates and even more so on cell-derived extracellular matrix (CDM), (ii) mitochondria are enriched at the protrusions of cells that are depleted of TRAK2, (iii) a 29-bp GA-rich motif in TRAK2 removes the protrusion-dependent scaling of TRAK2 mRNA and phenocopies the TRAK2 RNAi mitochondrial localization (iv) cells that have the GA-rich motif knockout have altered speeds of movement on CDM (but not glass) (v) localized translation of TRAK2 mRNA and association of the resulting TRAK2 with MIRO1 is likely the mechanism by which mitochondrial position is maintained as the cell changes shape.

While the underlying mechanism of cell-size-dependent mitochondrial scaling is an important premise, this manuscript fails to show conclusively that TRAK2 mRNA enrichment at the cell protrusions underpins mitochondrial positioning or function. I therefore cannot recommend the article to be published in its current form. My specific comments are:

1. The title alludes to mitochondrial function - however, there were no experiments conducted to test the

functional status of mitochondria. This would be essential to claim that mitochondrial function is changed in cells without TRAK2 mRNA enrichment at cell protrusions.

Author comment: The reviewer raises a key point that was critical to address with additional experiments. As such, we have probed mitochondria function upon staining of Wt and Δ TRAK2 cells with the cell-permeant dye, tetramethylrhodamine methyl ester (TMRM). TMRM is a fluorescent cation that accumulates in the mitochondrial matrix, with the extent of accumulation being tightly dependent on mitochondrial membrane potential. As such, TMRM is a well-established tool for assessing mitochondria functional status (doi: 10.1038/onc.2012.494; doi: 10.1016/S0006-3495(99)77214-0). Following staining of Wt and Δ TRAK2 cells with TMRM, we observed that the ratio of TMRM to mitochondria staining was significantly enhanced, specifically at the distal sites of cell protrusions (new Fig.5a-c). Thus, not only do mitochondria accumulate at distal regions of Δ TRAK2 cells, but these mitochondria also exhibit enhanced activity versus Wt mitochondria. Indeed, it is well-established that tight control of mitochondria transport by TRAK2-MIRO1 is critical to the spatial control of mitochondrial bioenergetics, especially in neurons (doi: 10.1038/nrn3156). Consistent with this concept, our new data indicate that TRAK2 mRNA-mediated control of mitochondria distribution likewise has a dramatic knock-on impact on the spatial control of mitochondria function in migrating cells. A full description of these new results has been added to page 12 Lines 8-18.

2. While the authors ruled out differences in protrusion shape/size as factors underlying accumulation of mitochondria at protrusions, the authors should also ensure that the underlying microtubule network is unchanged in control and TRAK2 siRNA/DelTRAK2 cells.

Author comment: As requested, we include new data confirming that the microtubule network is indeed unperturbed in siTRAK2 versus siCTRL cells (new Supplementary Fig.4a,b; page 7, lines 23-24), as well as Δ TRAK2 versus Wt cells (new Supplementary Fig.7a,b; page 11, lines 7-8).

3. It is unclear why mitochondrial accumulation at the protrusion tip would lead to increased velocity and movement of cells on CDM. The authors refer to previous work where mitochondria appear polarized to the leading edge in migrating cells, but the mitochondria in DelTRAK2/siRNA cells does not appear to be polarized. This part of the work needs to be developed extensively to make any claims about mitochondrial function being altered and thereby leading to increased motility of cells.

Author comment: As discussed in more detail above, we now include new data demonstrating that mitochondria membrane potential is significantly enhanced in mitochondria at the leading edge of migrating Δ TRAK2 cells (new Fig. 5a-c). Thus, these new observations confirm that perturbation of TRAK2 mRNA localisation indeed enhances local mitochondria activity at distal sites of protrusions. Moreover, it is also important to note that the increased accumulation of mitochondria that we observe at the leading edge of Δ TRAK2 cells (~2-fold) is highly comparable to the redistribution of mitochondria associated with increased cell motility in many other cell types (e.g. doi: 10.1039/D3NA00478C; doi: 10.1091/mbc.E16-05-0286; doi: 10.1073/pnas.150072211; doi: 10.1038/onc.2012.494). Thus, this distal accumulation is consistent with the observed knock-on increase in cell motility.

4. So too, the role of motor proteins esp. cytoplasmic dynein in mediating the proposed TRAK2-MIRO1 facilitated retrograde mitochondrial transport has not been shown and is key to the conclusions of the paper.

Author comment: We apologise for the lack of clarity on this issue, as there is over 20 years of literature that already confirms the fundamental and conserved role of cytoplasmic dynein in the control of TRAK2-MIRO1-mediated mitochondria retrograde transport (recently reviewed here, doi: 10.1002/pro.3839). As such, it is well established that perturbation of TRAK2 function will critically disrupt dynein-mediated retrograde transport of mitochondria. To fully clarify this point we have added reference to this extensive body of literature when discussing the importance of spatial control of TRAK2-MIRO1 interactions to dynein-mediated mitochondria transport, related to Fig.6 (page 13 line 25, and page 14, lines 1-2). Moreover, it is important to note that knockdown studies of cytoplasmic dynein would likely not be informative in the context of this study. Dynein loss-of-function has widespread impacts on many aspects of cell trafficking, dynamics and function, precluding specific assessment of its role in cell-size-scaling of mitochondria distribution. Consequently, the subtle manipulation of *TRAK2* mRNA localisation exploited in our study provides a much more elegant and targeted approach to define the function of dynein-TRAK2-MIRO1-regulated transport in the control of cell-size-dependent mitochondria localisation.

5. The number of PLA spots showing interaction between TRAK2 and MIRO1 (by the authors' own admission) is only 4-5, even in WT controls. Additionally, MIRO1(/2)-independent TRAK1/2-dependent mitochondrial transport has been established in MEFs (DOI: 10.15252/embj.201696380) - how important is the (loss of) TRAK2-MIRO1 interaction?

Author comment: The reviewer raises a key point about the relative importance of TRAK2-MIRO1 interactions in the retrograde transport of mitochondria, suggesting that MIRO1-independent mechanisms may alternatively direct TRAK2-mediated mitochondria transport. However, TRAK2-dependent 'retrograde' transport of mitochondria is well-established to be entirely MIRO1-dependent. For example, whilst the very interesting study cited by the reviewer reveals that TRAK2 can still associate with mitochondria and promote 'anterograde' transport in the absence of MIRO1, importantly, retrograde transport of mitochondria was indeed shown to be entirely MIRO1-dependent in this previous study. Thus, rather than contradicting our model, the cited work further confirms that TRAK2-MIRO1 interactions are fundamental to promoting retrograde mitochondria transport. However, as the reviewer astutely points out, we were also initially surprised by the low rate of TRAK2-MIRO1 interactions detected in migrating cells. However, the number of detected TRAK2-MIRO1 interactions is highly equivalent to the number of TRAK2-Mfn1 interactions previously observed using a similar proximity ligation approach (~2 to 4; doi: 10.15252/embj.201696380). Importantly, this small number of TRAK2-MIRO1 interactions observed is also likely a consequence of the recently documented highly transient nature of these interactions (doi: 10.1101/2021.06.03.446977). Thus, there is simply a low number of interactions available for detection at any one time due to their short-lived nature.

6. Fig. 2: The mitochondrial intensities in TRAK2 RNAi and delTRAK2 have been normalized to the corresponding segments of the protrusions in control cells. However the total mitochondrial intensity seems altered in TRAK2 RNAi/DelTRAK2- it would be ideal to quantify mitochondrial intensity as a proportion of the

total mitochondrial intensity *within* each cell so as to show true enrichment. So too, quantifying the mitochondrial intensity per unit volume of each segment is a better measure.

Author comment: As requested, we have quantified mitochondrial intensity at each position along a protrusion as a proportion of the total mitochondrial intensity for siCTRL/siTRAK2 cells (panel a below) and Wt/ Δ TRAK2 cells (panel b below). It is important to note that cells cultured in cell-derived matrix exhibit a stereotyped morphology that is highly elongated and tapered (see Fig.4d). Thus, the size of the distal protrusion is distinctly much smaller than the size of the proximal zone, and consequently a much smaller proportion of the total mitochondria are located at distal sites. Importantly, upon both siTRAK2-treatment (panel a) and Δ TRAK2 mutation (panel b), highly significant \sim 2-fold changes in mitochondria enrichment were detected specifically at the most distal sites. Thus, as expected, these fold changes mirror the fold changes observed in normalised quantifications displayed in both Fig.2f and Fig.4c. Yet, due to the dramatic differences in proximal versus distal mitochondria levels, these new graphs are much harder to interpret than the normalised data presented in Fig.2f and Fig.4c, therefore, we would prefer to retain the original graphs. However, these new analyses can be added to the manuscript if required.

Likewise, as requested, we have also quantified the mitochondria intensity per unit volume of each segment for Wt/ Δ TRAK2 cell (panel c below). These measures are highly technically challenging as the distal zone is very narrow and exceptionally thin (see Fig.4d). Thus, it is very difficult to robustly 3D render very thin cell volumes, both using IMARIS automated and manual rendering approaches. Despite this challenge, we have repeated Wt/ Δ TRAK2 experiments to achieve the best quality 3D rendering possible and, as expected, further confirm that mitochondria are significantly \sim 2-fold enriched at distal sites of Δ TRAK2 cell protrusions (panel c). Moreover, whereas a significant reduction in mitochondria per unit volume is observed at distal zones in Wt cells, in Δ TRAK2 cells mitochondria remain enriched at equivalent levels at all distal sites versus the proximal zone (panel c). Whilst these data fully confirm our previous observations of \sim 2-fold mitochondria enrichment 'per unit area' (Fig.4c), we consider this new analysis is not optimal due to the technical challenges of rendering these thin distal sites, creating significant noise in the data. Thus, we would prefer to focus on the more robust measure provide by mitochondria enrichment per unit area (presented in Fig.2f and Fig.4c). However, these new measures of mitochondria intensity per unit volume can also be included in the manuscript if requested.

7. The only part of the schematic that appears in Fig. 6i that has been demonstrated is the enrichment of TRAK2 mRNA at protrusions and perhaps TRAK2-MIRO1 interactions. The rest of the mechanism is speculative, and has to be demonstrated - via live cell imaging, photoactivation/photoconversion experiments of the mitochondria and by visualising/disrupting the motors in control and TRAK2 siRNA/DelTRAK2 cells.

Author comment: We are happy that the reviewer recognises the key novelty of this manuscript, i.e. that *TRAK2* mRNA localisation is fundamental to defining the site of TRAK2-MIRO1 interactions, which critically underpins cell-size-scaling of mitochondria positioning. Moreover, as the reviewer points out, we acknowledge that the proposed model in Fig.6i includes reference to the function of kinesin and dynein motor proteins that are not directly experimentally assessed in this manuscript. However, it is important to clarify that there is over 20 years of literature that has already conclusively confirmed the illustrated functions of kinesin/dynein and TRAK1/2. Thus, we argue that their function does not need to be re-assessed again in this study. Moreover, for reasons discussed below, experimental manipulation of these motor proteins would not be particularly informative in defining the key function of *TRAK2* mRNA in cell-size-scaling of mitochondria positioning – which is the core focus of this study. In particular, **kinesin-TRAK1-MIRO1** interactions are already a well-established and highly conserved critical driver of anterograde mitochondria transport (recently reviewed here, doi: 10.1002/pro.3839). Thus, distal delivery of mitochondria (and their accumulation in the absence of *TRAK2* mRNA localisation) will no doubt be largely kinesin-TRAK2-MIRO1-dependent, as illustrated in Fig.6i. However, even if other mechanisms do contribute to distal mitochondria delivery, this would not impact our proposed model, as it focuses on TRAK2 function after mitochondria have been delivered to distal sites (and how they get there does not matter). Moreover, experimental manipulation of kinesin/TRAK1 would likely not be informative in the context of this manuscript, as the resulting lack of distal mitochondria removes any pool of mitochondria for TRAK2 to act on. Regarding **dynein-TRAK2-MIRO1**, as mentioned in response to point 5, these interactions are again a very well-established and highly conserved critical driver of retrograde mitochondria transport (again, recently reviewed here, doi: 10.1002/pro.3839). Moreover, as discussed in response to point 4, the severe pleiotropic effects of dynein loss-of-function would preclude any informative assessment of the function of *TRAK2* mRNA in mitochondria positioning. Despite these points, we fully understand the reviewers concerns and apologise for failing to clarify the extensive body of literature supporting the model illustrated in Fig.6i. As such, we now include comprehensive referencing in the figure legend to bolster these points. Moreover, we now further clarify in the figure legend itself that this is a “hypothesised” model.

8. The data from independent experiments seem to have been pooled and the stats seem to have been performed on the number of cells. It would be good to indicate the different replicates (using diff colors) and to do the stats on the independent repeats (n=3 for instance, rather than n=no. of cells)

Author comment: As the referee correctly points out, data in this study has been presented as pooled cells from several independent experiments to enable visualisation of all underlying individual data points, and statistical tests applied to these pooled data sets. This is a standard approach applied in cell biological studies published in *Nature Communications* - as is evident from the most recent cell biology manuscripts published in *Nature Communications* in the past week (doi: 10.1038/s41467-025-58538-3; doi: 10.1038/s41467-025-57928-x; doi: 10.1038/s41467-025-58909-w; doi: 10.1038/s41467-025-58779-2).

9. The smFISH images need to be represented accurately - the contrast in these images seems to have been set at a level where only individual punctae are visible, but it would be essential to provide raw images where the background can also be seen. So too, some smFISH images which contain the red mRNA spots are not accompanied by the original images - this makes it very hard to actually see where the spots are.

Author comment: We would like to reassure the reviewer that images have not been set at a contrast level that obscures any background signal. Due to the multi-probe nature of smFISH, this approach offers exceptional signal to noise. Thus, highly fluorescent RNA spots are easily distinguishable, albeit small due to the single-molecule resolution of smFISH. Moreover, as discussed in response to reviewer 1, point 4, RNA spots were defined and counted in a non-biased automated manner using the well-established FindFoci ImageJ plugin (doi: 10.1371/journal.pone.0114749). As such, smFISH images presented in this study faithfully display only the computationally confirmed individual RNA spots identified by FindFoci. Whilst this a gold-standard approach in field of RNA localisation for both identifying RNA spots and displaying smFISH images, we understand the importance to confirm that the seemingly high-contrast images presented in this study do not obscure any important background details. Consequently, below we display raw data from the RNA-enriched zone of the first cell in Fig.1a, with the image brightness and contrast sequentially modified to reveal usually indiscernible background details. These images confirm that even upon high brightness / low contrast manipulation (right panel), the very bright RNA spots (red arrowheads) are again easily distinguishable and readily recognised by the FindFoci ImageJ plugin. Finally, as requested, for all images that were not originally accompanied by an unannotated original image (Fig.1a, d, g and Supplementary Fig.S3a,d), we have now added these to the manuscript (new Supplementary Fig.S3g,h).

10. Overall the methods need to be better described - how is a protrusion identified/defined? Where is the protrusion length measured from? The centroid of the nucleus? Each of the figures need to have extensive methods sections.

Author comment: As requested, we have added new text to the Methods section defining that motile cell protrusions were identified as the cell protrusion exhibiting either the highest enrichment of localised mRNAs, polarised perinuclear enrichment of mitochondria or enrichment of TRAK2-MIRO1 PLA foci. As cells were predominantly uniaxially elongated, motile protrusions were usually easily distinguishable as the longest protrusion extended by a cell (page 25 line 25-26, and page 26 lines 1-4). Moreover, we have included new Methods text confirming that the length of the protrusion was measured as the distance from the nuclear

envelope to the leading edge (page 26, lines 4-5). We hope that these new text additions (and other extensive methods text additions included in response to reviewer 1) provide sufficient clarity on the methods queried by the reviewer.

11. Some terms (e.g.: “FIMO analysis”) have been introduced without context which make parts of the manuscript hard to follow. So too, having the phrase “(data not shown)” is not ideal - surely there must be a way to show the data?

Author comment: As discussed in response to Reviewer 1, point 9, we have now added new text confirming that the ‘Find Individual Motif Analysis’ (FIMO) program is a search tool used for identifying known RNA/DNA motifs in any sequence of interest (Page 4, lines 18-21). Moreover, we have removed reference to “data not shown”, as this simply refers to a negative result from the FIMO analysis. As such, there is simply no data output to show for this analysis.